# Acquisition of *Escherichia coli* carrying extended-spectrum ß-lactamase and carbapenemase genes by hospitalised children with severe acute malnutrition in Niger

Kirsty Sands [1,2] ✉, Aditya Kumar Lankapalli[1], Giulia Lai[2], Brekhna Hassan[2], Edward AR Portal [1,2], Jordan AT Mathias[2], Ian Boostrom [2], Mei Li[1,2], Kate Cook[1], Shonnette Premchand-Branker[1], Lim S. Jones[3], Nathan Sayinzonga-Makombe[4], Sheila Isanaka[5,6], Rupa Kanapathipillai[7], Christopher Mambula[7], Isabelle Mouniaman[7], Céline Langendorf[5], Timothy R. Walsh[1,2] & Owen B. Spiller [2]

Hospitalisation and routine antibiotic treatment are recommended for children with complicated severe acute malnutrition (SAM) but this may exacerbate antimicrobial resistance. Here, we investigate carriage of Gram-negative bacteria in children under five years of age receiving treatment for SAM in Niger, comparing the frequency of colonisation with bacteria carrying resistance genes at admission, during hospital stay and at discharge. *E. coli* isolates carrying a $bla_{NDM-5}$ gene were selected for whole-genome sequencing. Rectal colonisation with bacteria carrying ß-lactamase genes is high, with 76% (n = 1042/1371) of children harbouring bacteria carrying a $bla_{CTXM-1}$-group gene and 25% (n = 338/1371) carrying a $bla_{NDM-5}$ gene. Over two-thirds of children who did not carry bacteria with a carbapenemase gene at admission are colonised with bacteria carrying a carbapenemase gene at discharge (n = 503/729, 69%). *E. coli* ST167 carrying $bla_{NDM-5}$ gene is recovered from 11% (n = 144/1371) of children. Here we highlight infection control and bacterial AMR transmission concerns amongst a vulnerable population in need of medical treatment.

A child with severe acute malnutrition (SAM) is defined by the World Health Organisation (WHO) as having low weight-for-height measurements according to WHO child growth standards, visible severe wasting, or nutritional oedema[1,2]. Malnutrition has a particularly high burden in low- and middle-income countries (LMICs)[3,4]. In 2020, the global prevalence of children with severe wasting was at least 13.6 million[5]. SAM is associated with a high mortality rate in children under five, often with co-morbidities such as HIV, sepsis, malaria, and tuberculosis[6–9].

Although SAM cases are often uncomplicated, between 10–30% require hospitalisation due to clinical complications. Diarrhoea and

[1]Ineos Oxford Institute for Antimicrobial Research, Department of Biology, University of Oxford, Oxford, UK. [2]Division of Infection and Immunity, Cardiff University, Cardiff, UK. [3]Public Health Wales Microbiology, University Hospital of Wales, Cardiff, UK. [4]Epicentre, Maradi, Niger. [5]Department of Research, Epicentre, Paris, France. [6]Department of Nutrition, Harvard T.H. Chan School of Public Health, Boston, Massachusetts, USA. [7]Médecins Sans Frontières, Paris, France. ✉e-mail: Kirsty.sands@biology.ox.ac.uk; sandsk1@cardiff.ac.uk

sepsis are among the leading causes of death in children with SAM, and clinical studies have predominantly identified *Enterobacterales* including *Escherichia coli*, *Klebsiella pneumoniae* and *Salmonella* species, in blood, stool, and urine samples[10–12].

Once a diagnosis of SAM is confirmed using precise anthropometric measurements[1], WHO guidelines recommend routine antibiotic administration[13,14]. Children with uncomplicated SAM (without signs of infection) are recommended a five-day course of oral amoxicillin[15,16]. In cases of complicated SAM parenteral or intravenous combination antibiotic therapy, such as benzylpenicillin, ampicillin/amoxicillin and gentamicin are recommended, with few reports of metronidazole usage[1,4,13]. SAM-associated infections are often successfully treated with amoxicillin/ampicillin and gentamicin in terms of antibiotic sensitivity and associated weight gain[16,17]. However, high rates of extended-spectrum β-lactamase (ESBL) carriage has been associated with amoxicillin exposure in outpatient settings for uncomplicated SAM treatment[18]. In addition, high rates of acquisition of ESBL-producing *E. coli* and *K. pneumoniae* have been reported in hospitalised SAM patients receiving broad-spectrum antibiotics, with a median hospital stay of 10 days[19]. Use of broad-spectrum antibiotics may select for intestinal carriage of multidrug-resistant (MDR) gut bacteria, predominantly *E. coli* and *K. pneumoniae*, potentially increasing the risk for transmission within and beyond the gut microbial community[18,19]. MDR is usually defined as non-susceptibility to at least one antibiotic in three or more antibiotic classes to which they were not intrinsically resistant[20].

Furthermore, healthcare facilities for children presenting with SAM are often overcrowded, with limited infrastructure and resources for adequate infection prevention and control (IPC) practices. These clinical environments pose a significant risk of bacterial transmission during medical procedures, patient-to-patient contact, and caregiver interactions, increasing the risk of nosocomial infection. There is limited data on the carriage of AMR bacteria in children presenting with SAM in Africa and particularly Niger. In this study, 103 patients presented with bacteraemia either on admission or during hospitalisation (admitted to the facility for medical treatment) and has been published previously[21]. Here, from the same cohort, we demonstrate a high prevalence of both rectal carriage and acquisition of ESBL and/or carbapenemase-producing Gram-negative bacteria (GNB) from children presenting with complicated SAM. Furthermore, we genomically characterise the dominant MDR *E. coli* clones circulating in the Madarounfa district Hospital, Maradi, Niger.

## Results
### Baseline characteristics
Between September 2016 and December 2017, 3004 rectal swabs were collected from 1371 children (aged 1–59 months; mean 17.9 months) suffering from SAM with medical complications at the Madarounfa district Hospital supported by Médecins Sans Frontières (MSF) in Maradi, Niger (Fig. 1, Source Data 1). The antibiotic most commonly prescribed to this cohort was ceftriaxone, administered both prior to and during hospitalisation as part of the MSF-supported nutritional treatment programme in Niger[21]. The cohort consisted of 761 males (1675 rectal swabs) and 610 females (1329 rectal swabs). The length of hospital stay ranged from 0 to 191 days, with 922 children (67%) hospitalised for 0–5 days, 366 (27%) for 6–10 days and 83 (6%) for more than 11 days. Two swabs were collected from most patients – one at admission and one at discharge (n = 1160, 85%). In addition, 264 swabs were collected during hospital stay (defined herein as 'hospitalisation swabs'), if clinical symptoms indicated that the patient was deteriorating, with 12 % (n = 168) of patients having three swabs in total, and 3% (n = 41) having four to seven swabs. For two patients, only a single swab was collected, in both cases at admission.

Overall, rectal microbiota from 1688 samples (1042 patients, 76%) carried a $bla_{CTX-M-1}$ group gene, 338 samples (301 patients, 22%) carried a $bla_{NDM}$ gene and 339 samples (296 patients, 21.6%) carried a $bla_{OXA-48}$-like genes. $bla_{KPC}$, the fourth antibiotic resistance gene (ARG) screened for, was not detected in any of the rectal swabs. Among these 1688 samples, 227 samples from 207 patients carried bacteria with both a $bla_{CTXM-1}$ group gene and a $bla_{NDM}$ gene, 189 samples from 178 patients carried bacteria with both a $bla_{CTXM-1}$ group gene and a $bla_{OXA-48}$-like gene, 91 samples from 89 patients carried bacteria with both a $bla_{NDM}$ gene and a $bla_{OXA-48}$-like gene, and, 51 samples from 51 patients carried bacteria with $bla_{CTXM-1}$ group, $bla_{NDM}$ and $bla_{OXA-48}$-like gene.

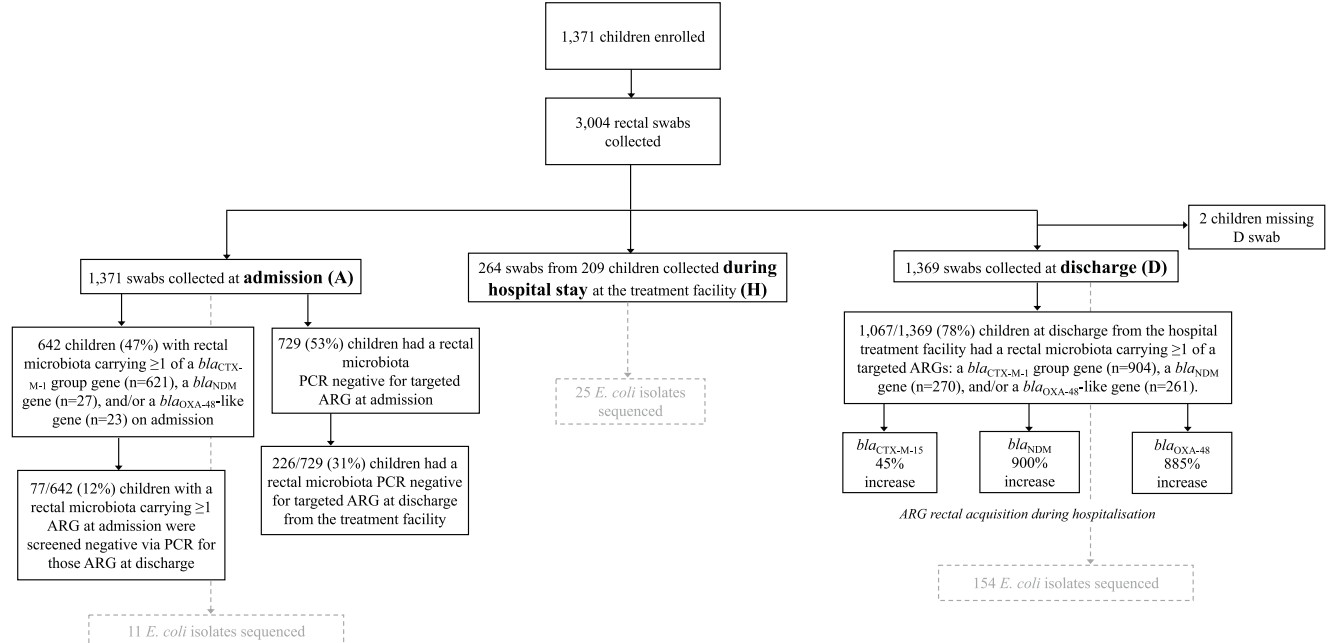

**Fig. 1 | Study flow diagram describing patient enrolment and rectal swab collection.** Rectal swabs were collected at admission (A), during hospitalisation if clinical symptoms suggested that the patient was deteriorating (H) and at patient discharge (D). The number of children with a rectal microbiota positive for screened antibiotic resistance genes (ARG) at admission and at discharge are summarised.

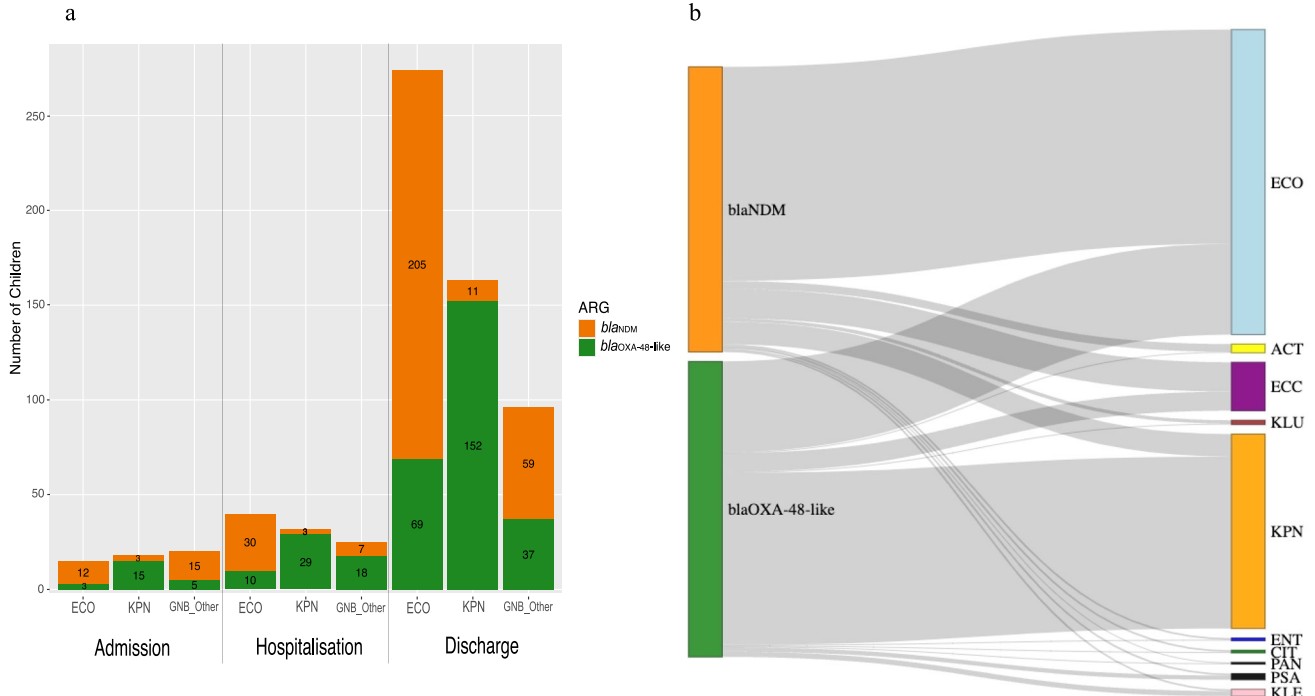

**Fig. 2 | Bacterial species carrying carbapenemase ARG in rectal microbiota.**
**a** The number of children who were carrying and acquiring bacteria with a carbapenemase gene in their rectal microbiota at admission, during hospitalisation for treatment, and at patient discharge. Data is split according to the two most dominant bacterial species; *E. coli* and *K. pneumoniae*, with the remaining grouped as others (a child with a/≥1 carbapenemase gene in different bacteria resulted in a single count per bacterial species). **b** The species diversity of bacterial isolates carrying a *bla*NDM and *bla*OXA-48 gene from rectal swabs; ECO – *Escherichia coli*, ACT – *Acinetobacter* spp., ECC – *Enterobacter cloacae* complex, KLU – *Kluverya georgiana*, KPN – *Klebsiella pneumoniae*, ENT – *Enterobacter* spp., CIT – *Citrobacter* spp., PAN – *Pantoea septica*, PSA – *Pseudomonas aeruginosa*, KLE – *Klebsiella* spp.

## Acquisition of Gram-negative bacteria carrying β-lactamase genes

At admission, 47% ($n = 642/1,371$) of patients tested positive for at least one of the screened ARGs, with 97% ($n = 621/642$) carrying a *bla*CTX-M-1 group gene (Fig. 1). Carriage of carbapenemase ARGs (*bla*NDM or *bla*OXA-48-like) at admission was low, with *bla*NDM genes detected in 2% ($n = 27/1371$) and *bla*OXA-48-like in 1.7% of cultured rectal microbiota ($n = 23/1371$).

By discharge, 78% ($n = 1067/1369$) of children tested positive for at least one of the screened ARGs (Fig. 1). The prevalence of *bla*CTX-M-1-like genes increased from 45% ($n = 621$) at admission to 66% ($n = 904$) at discharge, representing a 45% increase (Figs. 1, 2a). The acquisition of carbapenemase ARGs was similar for both *bla*NDM and *bla*OXA-48-like, with a 900% increase in *bla*NDM carriage (from 27 to 270 cases) and an 885% increase in *bla*OXA-48-like carriage (from 23 to 261 cases) from admission to discharge (Figs. 1, 2a). Among the children who carried bacteria with a *bla*NDM gene or a *bla*OXA-48-like gene at admission, not all remained colonised at discharge, with $n = 11/27$ (*bla*NDM) and $n = 14/23$ (*bla*OXA-48-like), children found to be negative for these genes at discharge.

## Bacterial diversity amongst carbapenemase ARG positive rectal carriage isolates

All rectal swabs in which at least one carbapenemase gene was detected were further scrutinised to identify bacterial species. In total, 673 bacterial isolates carrying a carbapenemase ARG were recovered, representing 15 different species across eight genera. The most common species carrying carbapenemase ARGs were *E. coli* and *K. pneumoniae* (Fig. 2a). Notably, *E. coli* positive for *bla*NDM and *K. pneumoniae* positive for a *bla*OXA-48-like gene were detected in the rectal microbiota from samples collected during hospitalisation and at discharge compared to those collected on admission (Fig. 2a). Of the 330 bacterial isolates from 348 patients positive for a *bla*NDM gene, most belonged to

*Enterobacterales*, with *E. coli* ($n = 248/364$) being the dominant species (68% of isolates). In addition, five different *Enterobacter* species carrying a *bla*NDM gene variant, along with *Citrobacter*, *Pantoea*, and *Pseudomonas* species, were identified (Fig. 2b). *Acinetobacter* spp. accounted for nine of the *bla*NDM-positive isolates. For the *bla*OXA-48-like-positive isolates, bacterial diversity was slightly lower, with 14 different GNB species identified (Fig. 2b) from a total of $n = 367$ isolates. While *E. coli* ($n = 105/367$; 28.6%) was among the top five bacterial species carrying a *bla*OXA-48-like gene, *K. pneumoniae* was the most dominant, representing 54% ($n = 199/367$) of cases. In 59/731 bacterial isolates, (35 carrying a *bla*NDM gene and 24 carrying a *bla*OXA-48-like gene), bacterial species identification via MALDI-TOF MS was unsuccessful after three independent tests.

## Genomic diversity of E. coli carrying a *bla*NDM gene

In total, 248 *E. coli* carrying a *bla*NDM gene were isolated from 3004 rectal swabs from 229 patients. Only twelve (5%, $n = 12/248$) were recovered from swabs collected on admission, with 12% ($n = 30/248$) recovered from swabs collected during hospitalisation. The remaining 83% ($n = 206/248$) were recovered from swabs collected at discharge from the treatment facility. Antimicrobial susceptibility testing was performed for 15 antibiotics on 217 (93%) of isolates (with 7% isolates PCR negative for *bla*NDM upon re-culture). Between 2019–2022 190 *E. coli* isolates carrying a *bla*NDM gene were sequenced ($n = 190/248$, 77% of the collection, Fig.1) to an average short read sequencing coverage of 30–70 X, median 53 X (Supplementary Fig. S1). The remaining *E. coli* isolates either could not be cultured or lost the *bla*NDM ARG/plasmid after recovery from −80 °C archive and were excluded. For all *E. coli* that were subject to WGS, *bla*NDM-5 gene was the only *bla*NDM gene variant. When a *bla*OXA-48 gene was identified, ($n = 2$), it was detected alongside a *bla*NDM-5 gene. Although 11 different MLST sequence types were identified, 148 (78%) isolates belonged to ST167

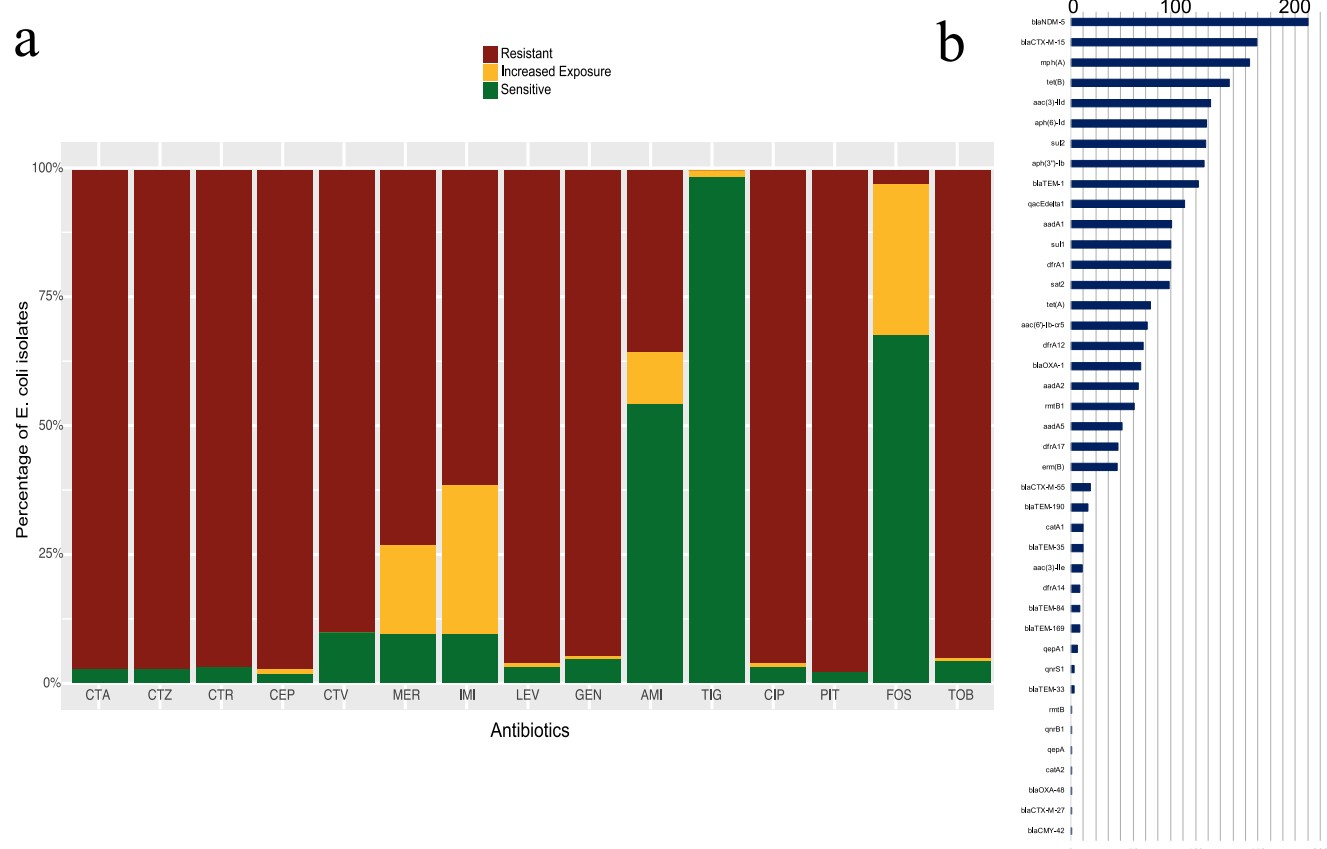

**Fig. 3 | Antibiotic susceptibility of *E. coli* isolates.** All isolates contain *bla*~NDM-5~.
**a** Stacked bar graphs showing the percentage of isolates with a sensitive, increased exposure or resistant profile to antibiotics; CTA – cefotaxime, CTZ – ceftazidime, CTR – ceftriaxone, CEP – cefepime, CTV – ceftazidime/avibactam, MER – meropenem, IMI – imipenem, LEV – levofloxacin, GEN – gentamicin, AMI – amikacin, TIG – tigecycline, CIP – ciprofloxacin, PIT – piperacillin/tazobactam, FOS – fosfomycin, TOB – tobramycin. **b** The number of *E. coli* genomes carrying antibiotic resistance genes.

and all harboured *bla*~NDM-5~. Other STs detected were *E. coli* ST1284 (*n* = 16, Supplementary Fig. S2, Source Data 2), ST11025 (*n* = 8, Supplementary Fig. S3, Source Data 3), ST448 (*n* = 5), ST2083 (*n* = 5), ST44 (*n* = 2), ST4977 (*n* = 2), ST443 (*n* = 1), ST8901 (*n* = 1), ST13159 (*n* = 1).

### AMR and genetic mechanisms of resistance in *E. coli*

Rates of resistance to carbapenems were high, as might be expected with 61% (*n* = 133/217) and 73% (*n* = 159/217) resistance for imipenem and meropenem, respectively. A further 17% (*n* = 38/220) had an intermediate MIC, and only 10% (*n* = 21/217) were fully sensitive to meropenem (Fig. 3a). Of the 21 isolates sensitive to meropenem, where WGS was available (*n* = 7/21), a carbapenemase gene was detected. Levels of AMR in *E. coli* carrying a *bla*~NDM-5~ gene were lowest to tigecycline (*n* = 0/217 0%), fosfomycin (*n* = 6/217, 3%) and amikacin (*n* = 77/217, 35%) (Fig. 3a). For the remaining 10 antibiotics tested including ceftazidime/avibactam, cefotaxime, ceftriaxone, gentamicin and piperacillin/tazobactam, resistance was >90% (Fig. 3a). 149 isolates had *bla*~CTX-M-15~, 16 isolates had *bla*~CTX-M-55~, one isolate had *bla*~CTX-M-27~, and 56 isolates had *bla*~OXA-1~ (Fig. 3b). At least one aminoglycoside resistance gene was identified in *n* = 187/190 isolates including *aac(3)-IId*, *aadA1* and *aph(6)-Id*, with 127 carrying a *tet*(B) gene, 64 carrying a *tet*(A) gene, 37 isolates carrying a *erm*(B) gene, and 11 carrying quinolone ARGs (*n* = 6 *qepA* *n* = 3 *qnrB1 and n* = 1 *qnrS1*; Fig. 3b).

### *E. coli* ST167 with *bla*~NDM-5~

Among 148 *E. coli* ST167 isolates recovered from 144 different children, eight (5%) were collected at admission, 19 (13%) during hospitalisation, and 120 (82%) at discharge. The high presence of *E. coli* ST167 carrying

*bla*~NDM-5~ at patient discharge from the treatment facility may cause concern for ongoing transmission as patients return home. The *E. coli* ST167 population were separated by ~ 450 SNPs (Fig. 4). One branch contains multiple small subclades (A–G) and one large subclade (H, 80 isolates collected December 2016-July 2017; ≤ 20 SNPs) and the other only a single clade (clade I ≤ 20 5 SNPs). The presence of multiple smaller subclades (clades A–G) alongside a larger subclade H (Fig. 4) may suggest evidence of local divergence. The most distant *E. coli* ST167 isolates (clade I, Fig. 4) were within five SNPs of each other, and all carried *bla*~CTX-M-55~, whereas the majority of *E. coli* ST167 (subclades A–H) carried *bla*~CTX-M-15~. In addition, isolates within clade I did not co-carry either *bla*~TEM-1~ or *bla*~OXA-1~ genes, unlike those in the other subclades (A–G; Fig. 4). Although multiple patients were colonised with a genetically similar strain of *E. coli* ST167 carrying a *bla*~NDM-5~ gene (subclades A–G, H), two co-occurring *E. coli* ST167 lineages were present in this cohort (clade I).

### *E. coli* plasmid population

The *E. coli* genomes contained multiple plasmid replicons, with IncF, Col and IncX3 being the most commonly detected via screening short-read genomes (Fig. 5a). IncF plasmid replicons were detected in 178 genomes with *n* = 106 isolates containing >1 IncF replicon (IncFIA, IncFII, IncFIB and IncFIC replicons detected, Fig. 5a), while IncX3 plasmid replicons were detected in 118 genomes. Long-read sequencing of 16 *E. coli* isolates positive for a *bla*~NDM-5~ gene revealed that *bla*~NDM-5~ was carried on two distinct plasmid types; IncX3 and IncF.

In *n* = 14/16 isolates (of different ST groups with available ONT data; *E. coli* ST167, *n* = 10; ST1284, *n* = 3, ST11025, *n* = 2, and ST2083,

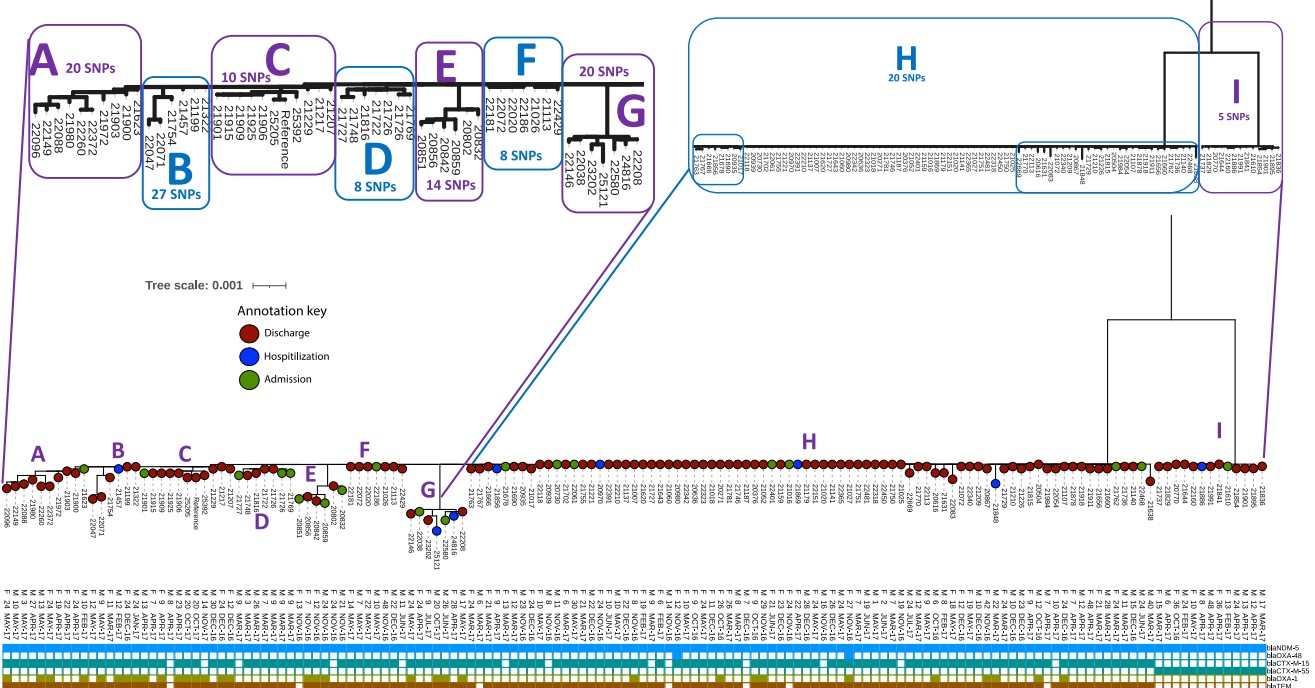

**Fig. 4 | E. coli ST167 SNP phylogenetic tree.** The clades within the tree are labelled A-I and are enhanced for visualisation in the purple and blue boxes. Blue boxes within clade H indicate isolates with the most pairwise SNPs detected within that group. The tree leaves are coloured according to sample type (at admission, during hospital stay, discharge) and named according to patient ID. The sex of the patient is indicated (F, female, M, male), followed by the age of the child in months and the date of the sample (month and year). The presence of acquired β-lactamase antibiotic resistance genes (ARG) is denoted with a presence/absence heatmap with a carbapenemase gene denoted blue, a $bla_{CTX-M-1}$ group denoted green and a $bla_{OXA-1}/bla_{TEM}$ (variants grouped) denoted brown.

$n = 1$; Fig. 5b and Supplementary Fig. S4). $bla_{NDM-5}$ was found on an IncX3 plasmid (46,161 bp) that is predicted to be conjugative (oriT, MOBP relaxase) using in silico methods. On the IncX3 plasmids ($n = 14$ E. coli isolates with IncX3 plasmids assembled), two prominent mutations were observed (Figs. 5c, 6), one upstream of the ATP-dependent metalloprotease gene (25,023 bp, nucleotide C to A) and at position 38,841 bp in a transcriptional regulator. The IncX3 plasmid containing the $bla_{NDM-5}$ gene mapped confidently to 132 E. coli isolates (Fig. 6), and 68 isolates carried both the mutations, seven isolates carried only 25,023 mutations and 28 isolates carried only the 38,841 mutation. This varying pattern of mutations among the samples indicates a heterogeneous mix of IncX3 plasmids circulating in the E. coli population with likely recombination or cross-transmission occurring. (Fig. 5b, Fig. 6, Source Data 5, Supplementary Fig. S4). PLSDB contains multiple hits of genetically similar IncX3 plasmids harbouring a $bla_{NDM-5}$ gene, highlighting the potential for transmission within and beyond the hospital setting.

Two E. coli ST167 isolates with long-read data carried $bla_{NDM-5}$ on a 132,316 bp plasmid with an IncFIA/FIC replicon that also carries $bla_{CTX-M-15}$, $bla_{OXA-1}$ and $bla_{TEM}$, and is predicted to be conjugative (oriT, MOBF relaxase; Figs. 5c, 6). As multiple IncF plasmids were often detected in the E. coli genomes (Fig. 5a) a larger variation of percentage read mapping and a higher proportion of SNPs were detected across the collection; however, in 56 isolates ($n = 53$ E. coli ST167, $n = 2$ E. coli ST11025, $n =$ E. coli ST1284), the mapping coverage was >98% (Fig. 6, Source Data 5), suggesting the presence of a genomically similar plasmid. It was not possible to confidently map two E. coli isolates from the $n = 190$ dataset (both ST167) to either the IncX3 (pIncX3-NDM5-or the IncF (pIncF-NDM5) plasmid sequences which may suggest either limitation with the depth of Illumina sequencing, or the presence of additional plasmid types harbouring a $bla_{NDM-5}$ genes may be present in the cohort that were not represented by the sub-cohort selected for ONT.

## Discussion

During the peak malnutrition period in 2019-2020 in Niger, it was estimated that up to one million children were acutely malnourished and required treatment, with approximately 400,000 diagnosed with SAM[22]. This prospective longitudinal study highlights the high prevalence and acquisition of β-lactamase genes during treatment for complicated SAM. Bacterial AMR is a major global health threat, with the highest associated mortality occurring in LMICs, as reported in the GRAM study in 2024[23]. Furthermore, the WHO classified third-generation cephalosporin-resistant *Enterobacterales* and carbapenem-resistant *Enterobacterales* as critical priority pathogens[24]. Our study found a high rate of rectal acquisition of the globally prevalent high-risk E. coli strain ST167 carrying $bla_{NDM-5}$ which was largely located on a globally disseminated and often highly conjugative IncX3 plasmid[25,26].

Malnutrition-induced alterations in the gut microbiota, particularly disruptions due to poor nutrition, enable the overgrowth of pathogenic bacteria[27–29]. Dysbiosis is thought to be exacerbated by the widespread administration of antibiotics, compounding the effects of nutritional deficiencies, including insufficient vitamins and minerals[30–32]. While broad-spectrum antibiotics are necessary for treating infections in children presenting with SAM, the risk of promoting resistance and adverse side effects must be carefully considered. A recent study by Mambula et al. (2023) found that among paediatric inpatients in Niger, the combination of ceftriaxone and gentamicin was most commonly prescribed (consistent with the findings of this study), while amoxicillin was the most frequently prescribed in the community[33].

Our data supports the recent study by Schwartz et al., 2023 which investigated the impact of amoxicillin on acute and long-term changes to the gut microbiome, in the same region[34]. Their findings showed an acute increase in antimicrobial resistance genes following seven-day amoxicillin treatment; however, these changes were not evident in the mature microbiome after two years of age. Similarly, in 2011, Woerther

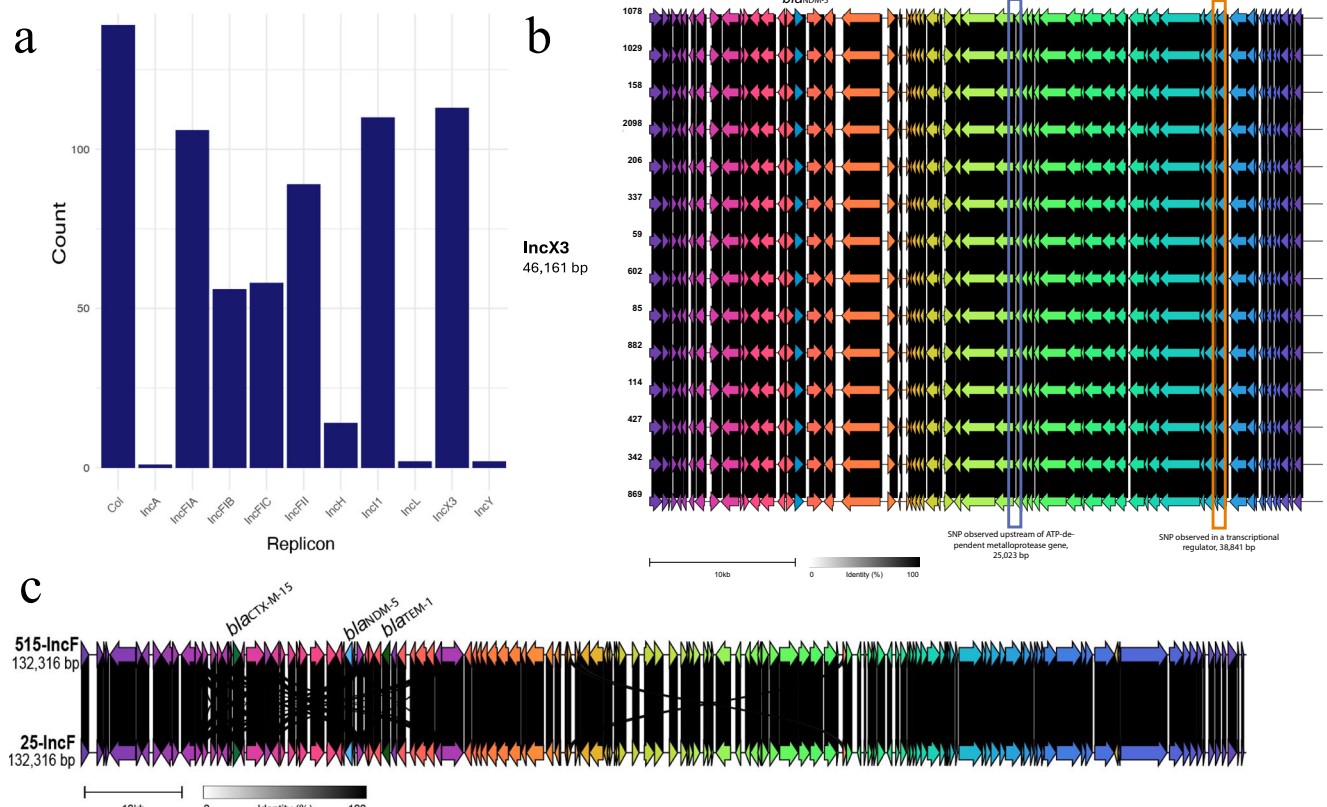

**Fig. 5 | _E. coli_ plasmid types. a** plasmid replicons detected across the $n = 190$ genomes grouped to 11 categories (**b**) plasmid genomic architecture of all 46,161 bp IncX3 plasmids carrying a $bla_{NDM-5}$ gene assembled from long read data, (**c**) plasmid genomic architecture of 132,316 bp IncFIA/FIC plasmid carrying a $bla_{NDM-5}$ gene

assembled from long read data. Individual plasmid sequences are annotated with the corresponding _E. coli_ isolate sample number (Source Data links sample number to study participant ID).

et al. reported a dramatic increase in the carriage of bacteria harbouring $bla_{CTX-M-15}$ in a SAM treatment facility in Niger[19]. $bla_{CTX-M}$ is the most clinically relevant and globally disseminated ESBL[35]. A decade later, our study confirms a high prevalence of bacteria carrying a $bla_{CTX-M-1}$ group gene, with an additional concern: the acquisition of MDR _Enterobacterales_ carrying $bla_{NDM-5}$. In this study we observed three predominant MDR _E. coli_ strains colonising children receiving treatment for SAM. MDR _E. coli_ colonisation was noticeably higher at discharge, suggesting healthcare associated acquisition and local transmission events. Previous longitudinal studies have shown that carriage with bacteria harbouring ARG is often transient[18,34].

It is particularly important to detect and understand the presence of antibiotic-resistant Gram-negative bacteria that may be transiently colonising the gut microbiota of vulnerable children, as this may pose a risk to subsequent infection[36,37]. In addition to the widespread circulation of _E. coli_ ST167 carrying $bla_{NDM-5}$, we identified other _E. coli_ ST carrying a $bla_{NDM-5}$ gene on different IncX3 plasmids, indicating multiple strains circulating among immunocompromised children. AMR plasmids may spread to other _Enterobacterales_ colonising the gut, while healthcare-associated acquired strains could spread into the community, potentially turning these treatment centres into hotspots for the dissemination of carbapenem-resistant _Enterobacterales_.

In addition, for 31 children in this study carrying _Enterobacterales_ with $bla_{NDM}$ in their rectal microbiota they were further colonised with _Enterobacterales_ species in the bloodstream[21]. As reported by Andersen et al., bloodstream infections in this cohort were predominantly caused by non-typhoid _Salmonella_; however, _E. coli_ was the second most common pathogen[21]. As part of standard care, children presenting with complicated SAM are hospitalised for treatment, including antibiotics, until stable. However, balancing appropriate antibiotic

use and preventing the selection and spread of MDR pathogens in low resource settings remains challenging.

We acknowledge several limitations in this study. Firstly, rectal swabs were collected during hospitalisation only in case of clinical deterioration evaluated after 48 h of hospitalisation. These specific samples represent the most vulnerable subset of included children. Secondly, rectal swabs were screened using targeted PCR for a specific ARG panel, limiting the assessment of the full resistome. Thirdly, the presence of multiple $bla_{CTX-M}$ variants in WGS data suggests that additional ESBL genes and variants were present but were not detected via PCR. Fourthly, Due to logistical and financial reasons, we were also unable to genomically characterise _K. pneumoniae_ isolates carrying $bla_{OXA-48}$-like genes. As all children receive antibiotics as mandated by guidance, we were not able to evaluate associations between AMR bacteria acquisition, antibiotic use, or to differentiate other contributing factors such as IPC.

Access to safe water, sanitation, and hygiene (WASH) is vital to prevent the spread of AMR bacteria. As IPC is intrinsically linked to WASH, this study underscores the urgent need to strengthen IPC measures in LMIC treatment facilities. Evidence-based IPC programmes should be reviewed and prioritised in inpatient settings, particularly for immunocompromised populations such as children presenting with complicated SAM. Determining whether correlations exist between bacterial acquisition and antibiotic therapy, medical procedures, and hygiene practices could guide treatment guidelines in reconsidering antibiotic treatment protocols for complicated SAM to mitigate the spread of AMR. Our study highlights the very serious issue regarding the dissemination _E. coli_ carrying $bla_{NDM-5}$ in Madarounfa District Hospital, Maradi. Unfortunately, _Enterobacterales_ carrying a carbapenemase gene is likely apparent in similar healthcare settings

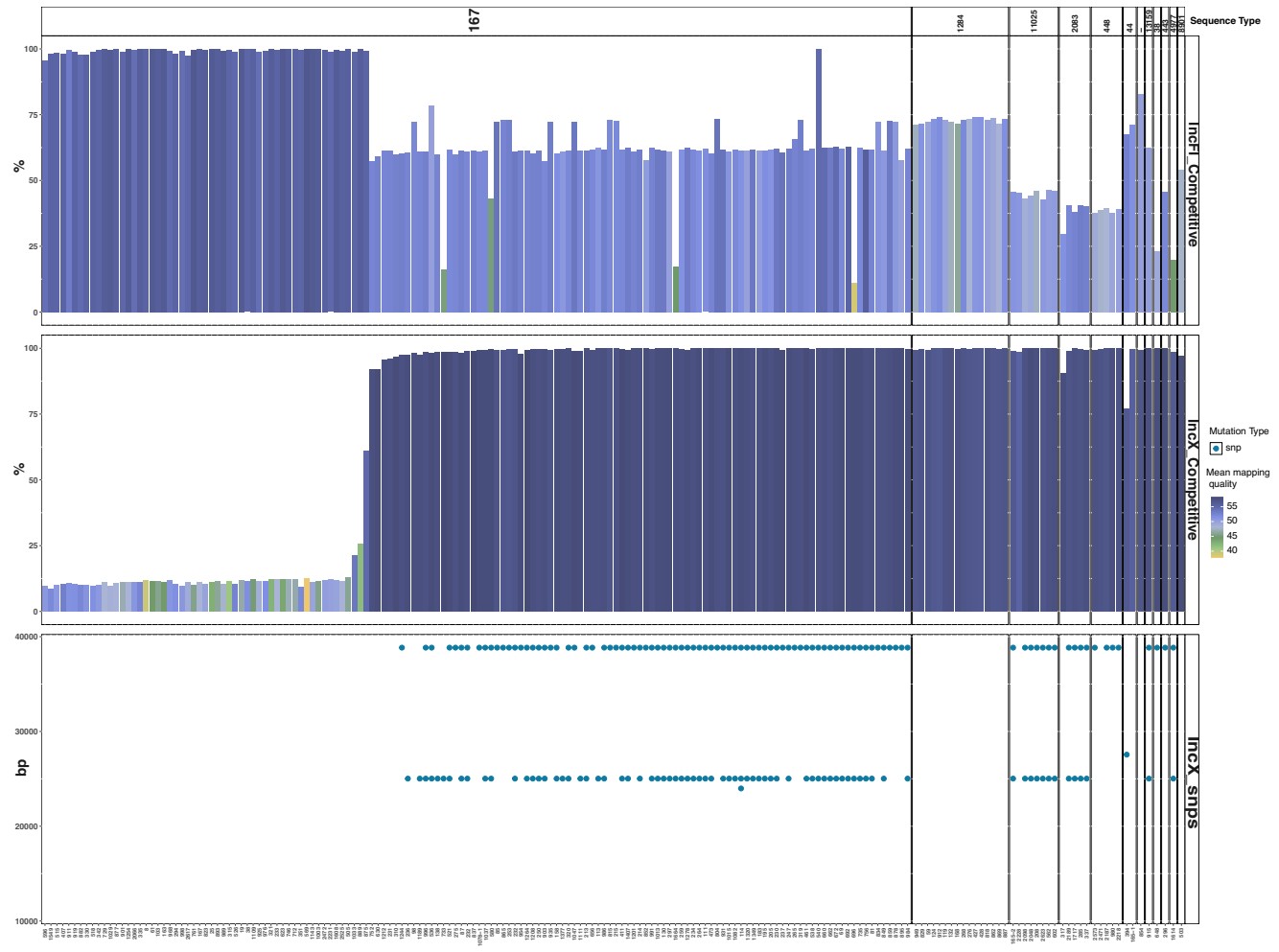

**Fig. 6 | *E. coli* plasmid population mapping.** Competitive *E. coli* short read mapping to the IncF *bla*$_{NDM-5}$ plasmid (pIncF-NDM5) and the IncX3 plasmid (pIncX3-NDM5-a), separated per *E. coli* ST. The prominent mutations observed in the mutations denoted in the IncX3 plasmid sequences across the *E. coli* population are also shown; snp – single nucleotide polymorphism.

across central and sub-Saharan Africa. As AMR continues to increase globally, concurrent humanitarian crises, including droughts and conflicts, are exacerbating malnutrition, leading to overcrowded treatment centres. While access to medical care is crucial, the conditions within these facilities and the prophylactic use of antibiotics may contribute to the emergence and amplification of AMR hotspots, posing a severe threat to the most vulnerable populations.

## Methods

### Study design

A descriptive and longitudinal study was conducted between September 2016 and December 2017 in Madarounfa, Maradi, Niger, at the Madarounfa Intensive Nutritional Rehabilitation Centre (CRENI) managed by the Niger Ministry of Public Health with support from Médecins Sans Frontières. Microbiology processing of rectal swabs was conducted in 2018, and the minimum inhibitory concentration and whole genome sequencing of *E. coli* was performed between 2019 and 2021. Children admitted to the CRENI between 0 and 59 months of age who did not require immediate resuscitation on admission were enroled[21]. Rectal swabs were collected according to three categories; (1) upon admission to the treatment facility (2) during inpatient stay, additional rectal swabs (defined as "hospitalisation swabs") were taken if clinical symptoms suggested that the patient was deteriorating, (3) at the time of patient discharge from the treatment facility. All rectal

swabs were collected with Amies plus charcoal media (COPAN Diagnostics)) and according to standard anorectal specimen collection guidelines. Upon admission and prior to enrolment in the study, informed consent was obtained from parents/primary carer. The ethical evaluation study was approved in 2016 by the National Consultative Ethics Committee of Niger (N007/2016/CCNE) and the Committee for the Protection of Persons, Ile-de-France (16007, Epicentre 2016). All ethical approvals were relevant for the study and permitted the enrolment of children, collection of rectal samples and analysis for research purposes. No statistical method was used to predetermine the sample size. No data were excluded from the analyses. The experiments were not randomised, and the Investigators were not blinded to allocation during experiments and outcome assessment.

### Processing of rectal swabs

Rectal swabs in preservative media were stored at +4 °C until transfer to the microbiology laboratory at Cardiff University, UK. Rectal swabs were streaked on three chromogenic agar media, Chromatic Detection (Liofilchem®, Italy) supplemented with either vancomycin (10 mg/L), vancomycin plus cefotaxime (VC, 10 mg/L and 1 mg/L), or vancomycin plus ertapenem (VE, 10 mg/L and 2 mg/L,) to select for GNB, cefotaxime resistant GNB (indicative of the presence of ESBL producers) and ertapenem resistant GNB (indicative of the presence of

carbapenemase producers)[38]. The $bla_{CTX-M-1}$ group was selected as the target ESBL to screen for in this study due to reported high prevalence and global dissemination[35]. $bla_{NDM}$ and $bla_{OXA-48}$-like ARG were screened for as they are often reported as the most prevalent carbapenemase genes in Africa[39,40]. Although the $bla_{KPC}$ carbapenemase gene is not endemic to Africa[40], it was also screened for in this study as part of a multiplex PCR. The presence of a $bla_{CTX-M-1}$ group gene was determined by PCR (F-ATGCGCAAACGGCGGACGTA, R-CCCGTTGGC TGTCGCCCAAT), and the presence of $bla_{NDM}$ (F-AGCTGAGCACCG-CATT, R-CTCAGTGTCGGCATCAC), $bla_{KPC}$ (F-TAGTTCTGCTGTCTTGT CTC, R-CCGTCATGCCTGTTGTC) and $bla_{OXA-48}$-like genes (F-GGCGTA GTTGTGCTCTG, R-AAGACTTGGTGTTCATCCTT) was determined by multiplex-PCR using the Illustra PuReTaq Ready-To-Go PCR Beads (GE Healthcare, USA), with all conditions described previously[41]. All bacterial cultures were preserved with TS/72 beads (Technical Service Consultants, UK) at -80 °C. All phenotypically distinct bacterial colonies were purified from samples positive for a $bla_{NDM}$ and/or $bla_{OXA-48}$-like gene for repeat multiplex-PCR. The bacterial species for isolates positive for any of the carbapenemase genes in the study were identified by MALDI-TOF MS (Bruker Daltonik GmbH, Coventry, UK). Due to the high prevalence of the $bla_{CTX-M-1}$ group in and the likelihood of multiple *Enterobacterales* species carrying a $bla_{CTX-M-1}$ group gene, we did not proceed to isolate bacterial species. The minimum inhibitory concentration of 15 antibiotics were determined for 217 isolates via agar dilution[38] and interpreted according to the European Committee on Antimicrobial Susceptibility Testing breakpoints (EUCAST, v13)[42].

### *E. coli* whole genome sequencing

Briefly, single colonies were placed into 1.8 mL of LB broth and incubated overnight at 37 °C. Total gDNA was extracted from pelleted bacteria using a QIAamp DNA kit (Qiagen) on a QIAcube instrument (Qiagen, Germany) with an additional RNAse step and quantified using the Qubit (v3.0) (ThermoFisher, USA). DNA libraries were prepared for paired-end sequencing using the Nextera XT v2 and sequenced using the V3 chemistry (300 bp x 2) on an Illumina MiSeq (Illumina, USA). Representative isolates were subject to long-read sequencing on the MinION (Oxford Nanopore Technology (ONT) to allow the hybrid assembly to produce higher quality genomes for variant mapping. Genomic libraries were generated using the 96-Rapid Barcoding Kit (SQK-RBK110.96; ONT) sequenced using R9.4 flow cells and basecalled using Guppy (v4.0.14) within MinKnow. Repeat ONT sequencing in 2024 was performed on 20 isolates to perform plasmid analysis. The 96-Rapid Barcoding Kit was used as described above, and *E. coli* genomes were sequenced using R10.4.1 flow cells using a GridION with base-calling performed using Dorado (Nov. 2024) within MinKnow.

### Short read bacterial assembly and genomic relatedness

Short reads were trimmed with trimgalore (--paired --phred33 -q 25 --nextera -e 0.2)[43] and the resulting reads assessed using fastQC[44]. De novo genome assembly was performed using Shovill (v0.9.0)[45]. Seqtk (v1.3)[46] was used to remove contigs less than 200 base pairs (seq -L 200) before submission to NCBI. Quast (v5.2.0)[47] was used to assess the quality of genome assemblies. Genomes with a genome length >6MB/ >1000 contigs were excluded. Trimmed paired-end short reads were mapped to the resulting de novo genome to assess sequencing coverage and depth parameters using bbtools/39.01 and bbmap[48]. Species identification was confirmed using Kraken (v2.0.8-beta)[49]. Short-read genomes were annotated using Prokka (v1.14.5)[50]. Multi-locus sequence types (MLST) were determined in silico using mlst (v2.22.0)[51]. Genomes not assigned to a known ST were uploaded to Enterobase[52] for the assignment of novel allele profiles. Contigs were screened for ARG using AMRfinderplus (v3.12.8)[53]. Plasmid replicons were determined using PlasmidFinder within ABRicate[54].

SNP phylogenies were constructed using high-quality (hybrid short-read and long-read genomes) reference genomes selected per ST of interest; ST167 (reference isolate 2525, genome accession JBMWUF000000000), ST1284 (reference isolate 869, JBMWUG000000000) and ST11025 (reference isolate 2623, JBMWUH000000000). Illumina fastq files were trimmed as described above, ONT sequence reads were trimmed using filtong (v0.2.1)[55] and hybrid assemblies were produced using unicycler (v0.4.9)[56]. Snippy (v4.6.0)[57] at default parameters was used to call variants against sequence reads, and to generate a SNP phylogenetic alignment of all SNP sites present in the samples. Snippy-clean prepared the resulting SNP alignment for input to recombination removal and phylogenetic tree construction. Gubbins (v2.3.4)[58] was used to remove possible recombination sites, and snp-sites (v2.5.1)[57] was used to extract SNPs from the multi-fasta alignment. A pairwise SNP matrix was created using Pairsnp (v.0.0.7)[59]. IQtree v2 was used to create the phylogenetic tree as outlined above. Figures were produced using R and packages *ggplot2* and *networkD3*.

### $bla_{NDM-5}$ plasmid analysis

Long reads of 16 samples ($n = 10$ ST167, $n = 3$ ST1284, $n = 2$ ST11025, $n = 1$ ST2083; Source Data 6) were assembled using Flye (v2.9.4-b1799) and polished with medaka (v1.12.1) after removing poor quality reads (Q12/Q15 and 1000 bp length) with nanoq (v0.10.0). Assembled contigs were reoriented (dnaapler v0.8.0), annotated (bakta v1.9.4 database v5.1), profiled for plasmid (mob_suite v3.1.9) and AMR genes (Abricate v1.0.1 and AMRfinderPlus v4.0.19 database v4.0) and screened for $bla_{NDM}$ genes among them. Plasmid sequences were uploaded to PLDSB (v2024_05_31_v2)[60] for comparative analysis to public repositories using mash (mash screen and mash dist). Out of the 21 samples, the $bla_{NDM-5}$ gene was found on 18 IncX3 (size: 46,161 bp) and 2 IncF (size: 132,316 bp) plasmids, respectively. Representative IncX3 and IncF plasmids from long-read assembled genomes harbouring $bla_{NDM-5}$ concatenated to form a single file of reference plasmids. Competitive mapping against a combined plasmid reference sequence (to improve mapping accuracy to either location) using bwa and variant detection of short read sequences from 190 samples against the combined plasmid reference was performed using snippy (v4.6.0). Samtools (v1.18) recorded the coverage of reads against the combined plasmid sequence as well as individual plasmid sequences. These coverage stats were compared along with the mutations for each plasmid. Gene synteny alignment of IncX3 and IncF plasmids were created for the representative plasmids carrying or missing $bla_{NDM-5}$ gene using clinker (v 0.0.31).

### Reporting summary

Further information on research design is available in the Nature Portfolio Reporting Summary linked to this article.

## Data availability

The *E. coli* Illumina sequence reads, Illumina genomes, R9.4 and Illumina hybrid genomes used for variant calling, R10.4 whole genome assemblies, and R10.4 extracted plasmid sequences used for mapping generated in this study have been deposited in the NCBI repository under the project accession PRJNA1096457. Source Data 5 and Source Data 6 specify all individual accession numbers. Source data are provided in this paper.

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

## Acknowledgements

This study was funded by Médecins Sans Frontières - Operational Centre Paris. We would like to thank Ali Aboklaish and Emma Kerr for their support in the laboratory processing for this study. We thank and acknowledge the hard work of the team at Liofilchem®, Roseto, Italy, specifically F. Brocco and M. Collett. We would like to acknowledge the Specialist Antimicrobial Chemotherapy Unit (SACU) at Public Health Wales for support with bacterial isolate identification using MALDI-TOF MS. We would also like to acknowledge Wales Gene Park and ARCCA for their continued bioinformatics support. Bioinformatics analysis was undertaken using the supercomputing facilities at Cardiff University operated by Advanced Research Computing at Cardiff (ARCCA) and we acknowledge its support via the Welsh Government through the Higher Education Funding Council for Wales (HEFCW). The Centre for Trials Research is funded by Health and Care Research Wales and Cancer Research UK. We thank the team of curators for the databases hosted on Enterobase. The authors would like to acknowledge the use of the University of Oxford Advanced Research Computing (ARC) facility in carrying out this work. KS has been funded by the Ineos Oxford Institute for Antimicrobial Research since 2021.

## Author contributions

T.R.W. and O.B.S. were the principal investigators of the study. T.R.W., O.B.S. and L.J. designed the study. K.S. draughted the manuscript. K.S., K.C. and S.P.B. performed literature searches. G.L., B.H., K.S., E.A.R.P., K.C., S.P.B. and M.L. performed the microbiology experiments. K.S., B.H., E.A.R.P., J.M. and I.B. performed the sequencing experiments. K.S. performed the microbiology data analysis and produced the figures. K.S. and A.L. performed the bioinformatics analysis and produced the figures. C.L., R.K., C.M., I.M. and S.I. designed the trial and patient study protocol. C.L., S.I. and N.S.M. enroled the patients and collected the samples. All authors contributed to data interpretation. All the authors critically reviewed and approved the final version of the manuscript. All authors had full access to the data in the study and had final responsibility for the decision to submit for publication.

## Competing interests
The authors declare no competing interests.
