## [Transparent Peer Review file · Nature Communications]

Acquisition of *Escherichia coli* carrying extended-spectrum β -lactamase and carbapenemase genes by hospitalised children with severe acute malnutrition in Niger

Corresponding Author: Dr Kirsty Sands

Version 0:

Reviewer comments:

Reviewer #1

(Remarks to the Author)

This manuscript describes screening of rectal swabs from infants with severe acute malnutrition (SAM), collected on admission, during hospitalization and on discharge from a single inpatient SAM treatment centre in Niger 2016-17. Isolates underwent susceptibility testing and PCR to detect one ESBL gene type and three carbapenemase gene types. Isolates with a carbapenemase gene underwent short read sequencing. Plasmids carrying blaNDM-5 in three isolates where long read data was available were analyzed, but there are also problems with the accuracy of the plasmid sequences and with the analysis. The choice these particular plasmids also appears to be opportunistic and it is not clear that they are particularly representative, as information about plasmid replicons etc in other isolates in the collection is not reported.

Although detecting increasing carriage of carbapenemase genes in bacteria from SAM patient might be important, some of the papers cited already show acquisition of ESBL genes by SAM patients in hospital (e.g. Lines 262-4, Ref 19), and the isolates examined here are now quite old (from 2016-17). The point of including other aspects is not always clear and leads to a lack of coherence.

The manuscript is not well presented, particularly the Supplementary Information, nor well written, and lacks attention to detail/accuracy. Lists of numbers in the text might be better just dealt with in figures, with only the main conclusions stated clearly. The manuscript is also wordy and too long for the amount of data presented. Figures and tables across both the main text and Supplementary are not well thought out and information is unnecessarily duplicated across text, tables and figures.

Main scientific points

1) It is not clear why only these four gene types (Line 342) were chosen for PCR and Results (Line 98) should state which gene types were screened for, not just those that were found.

The "blaCTX-M-15" primers (Supplementary Table 7) would also detect other blaCTX-M-1 group genes (maybe also including blaCTX-M-1 itself, though it has one mismatch in the reverse primer) that could not be distinguished without sequencing. This must be explained much more accurately.

Line 101 – "ARG gene types" would be more accurate, as variants are not defined by PCR.

Line 176 mentions that sequencing detected blaCTX-M-55, a variant of blaCTX-M-15 that also has 100% match to both "blaCTX-M-15" primers, and blaCTX-M-27, which would not have been detected by these primers (as it belongs to the blaCTX-M-9 group) and so could be present in other isolates.

2) Sequence analysis (Supplementary Figures 3, 4, 6 and associated text).

The sequence analysis has problems, there are inaccuracies and the text below each of these supplementary figures reads like lab notes, not part of a finished manuscript.

Figs S3 and S4

Alignment of the IncFII plasmid sequences from isolates 889 and 2505 provided suggests that most differences between them could be errors (e.g., differences in number of the same base in homopolymer regions, indels in IS with well-defined sequences, some of which would break transposase genes). All differences need to be checked e.g. by mapping raw reads, before sequences are compared to identify only differences that are real.

Fig. S3

It's not clear what this is showing – it doesn't seem to be the whole plasmid?

The name "pGL2525" is used but not explained.

The ARG gene names are not correctly formatted, differ between the figure and legend and could be written in a more logical order in the legend e.g. the *dfrA12-gcuF-aadA2* is a cassette array followed by *sul1*, Tn3 might actually be Tn2, which is more common, TnAs1 might be Tn21? Tn5403 is mentioned in the legend but not shown in the figure, "ItrA group II intron reverse transcriptase/maturase protein" should just be referred to as "group II Intron" (see Zimmerly Lab for Mobile Group II Introns (ucalgary.ca)),

"tra operon" or "tra gene cluster" would be better than "cassette".

"dhfr" is incorrect – its *dfrA12* as shown in the figure

Which cat gene? – it's probably *catB3delta*? And it should be *blaOXA-1*.

IS need to be properly identified by name or number (see ISFinder, <https://www-is.biotoul.fr>)

The region shown is not a "cassette".

% homology is meaningless – homology is not quantifiable.

The relevance of the results of the comparative analysis is not explained.

Lines 207 – both fully sequenced plasmids are from clade C so maybe it's not surprising that they are probably identical.

Line 209 – "motility" is the wrong word here. "predicted to be conjugative".

Line 208 – what is meant by FIA/C -FIA-FC? i.e. FIA+FC type replicons? FIA-FII is used elsewhere.

Line 211 - "10 hits contained very similar ...plasmids" does not make sense.

There is no exploration of whether the short reads from the other isolates in Clade C or other clades could be compatible with having the same plasmid e.g. same ARGs sets, mapping etc.

Fig. S4

This map probably corresponds to the plasmids from both 889 and 2525?

The text is hard to read and is not clear why some things are labelled and some not – all of the ARG shown in Fig. S3 are present as well and the relevance of labelling various RNA is not clear.

The 1.1 kb insert seems to be IS903 (see ISFinder).

Including a quote from a paper is not appropriate – the information needs to be properly related to the data obtained here.

Fig. S6

bleMBL is so named because of its known association with *blaNDM* and these genes are almost always found together.

The elements upstream of *blaNDM-5* are a fragment of *ISAb125*, well known to be associated with *blaNDM* genes in many different contexts, derived from Tn125, and here interrupted by *ISKpn26*. Both of these IS are IS5 family and the extents and names of IS need to be properly identified using ISfinder

The meaning/purpose of including most of the other text here is unclear. This plasmid seems to match a well-known *IncX3* plasmid carrying *blaNDM-5* (and some other variants), so discussing a comparison with only a few other plasmids is unhelpful. Is 99% identity meant? This should be stated. Using "NDM cassette" to mean the immediate context of *blaNDM* should be avoided, due to possible confusion with gene cassettes associated with integrons. Again, this plasmid sequence should be carefully checked for errors and corrected if necessary.

3) Other scientific points

Line 35 – isolates of the same ST would be expected to be "genomically similar"?

Line 66 – what is the definition of multidrug resistant?

Line 80 – what type of average? The mean? This should be stated.

Line 119 – as rectal swabs were taken shouldn't it also be "rectal carriage here?"

Lines 114-20 – did any children with bacteria with a carbapenemase genes on admission have no such bacteria on discharge?

Line 161 – why is a range given here? This needs to be explained. Is "a few isolates were negative for a *blaNDM* gene on repeat PCR" meant here? "few isolates" has a different meaning.

Line 169-71 – are these 21 of the 248 *E. coli* with *blaNDM* detected by PCR stated at the beginning of this section? Are these 7 isolates part of the 190 that underwent WGS? They all had *blaNDM-5*? What could be a possible explanation for the genotype/phenotype discrepancy?

Line 179 – which *qnrB* gene(s)? Why 183 with WGS here and 190 elsewhere?

Line 445 – BioProject PRJNA1096457 is now available, but no complete assemblies are listed and at least the three complete (and checked) plasmid sequences should be included/submitted separately to GenBank and the accession nos. listed here.

4) Discussion

This mainly refers to previously published information and does not make it clear how the current study adds to this.

Lines 243-4 - Results only cover two ST167 isolates with an F plasmid with *blaNDM-5* and one ST1284 isolate with *blaNDM-5* on an *IncX3* plasmid. No information on plasmids in other isolates is mentioned in Results. How is it known that the *IncX3* plasmid is "highly conjugative"?

Lines 265-6 – could plasmids be being acquired by resident *E. coli*?

Lines 268, 273 – ST rather than strains? Different strains can be the same ST. Also "the ST167 global strain".

Line 272 – what is the evidence for this "dominant circulation" of the *IncF* plasmid?

Line 274 – where is this information in Results?

Line 288 – "elucidate" is not the best word here.

Line 292 – there is nothing in the Methods or Results about how this deterioration was evaluated.

Line 293 – wording problems.

Line 309 – *blaNDM-5*?

5) Methods

Line 322-4 – how were rectal swabs taken? The information on Lines 322 is not needed here, as it is in subsequent

sections.

Lines 326-7 – blood samples were not examined as part of this study?

Line 328 – what is meant by “time of deterioration”?

Line 335 – specify the media.

Line 342 – state which was done by PCR, and which were done by multiplex PCR.

Line 346 - were any samples positive for blaKPC? How many tested positive then negative? Was retesting carried out immediately or after storage?

Lines 356-7 – the text in the results implies that *E. coli* from rectal swabs, not blood cultures, were sequenced. Which is correct?

Line 368-423 - this section is oddly structured with section headings and overlapping numbered sections.

References/website are needed for the various software used.

Line 390 – numbering of bla genes is from protein sequence. ResFinder can be inaccurate as it is based on nucleotide sequence searches.

Line 391 – PlasmidFinder really finds replicon types. The PlasmidFinder results are only presented for the 3 isolates with long read sequencing.

Line 411 – what are these references? Are the sequences available?

6) Figures

None of the legends are worded well, e.g. there is usually no need to describe the type of figure. See comments above about correctly describing what the primers will actually detect etc.

Fig. 1

This is not particularly easy to follow and the information is also in the text. A clearer figure (supplementary?) would mean that most of the detail could be removed from the text.

Fig. 2

The meaning of “Deterioration” here is not clear. The legend refers to “Admission, hospitalization (during patient admission) and discharge”, which is not clear either. There is no description of part (b) and the same information seems to be shown in more detail in Fig. 3(b) and (c).

See comments above about correctly describing what the primers will actually detect.

Fig. 3

(a) see comments on Fig. 1

(b) and (c) duplicate Fig. 2b.

Legend – all isolates in the study were “purified from rectal samples”?

Line 152 – “species” not “isolates”?

Fig. 4

blaNDM-5 does not need to be shown if it is present in all isolates – just state in the title.

The order Admission, Hospitalization, Discharge in the key would be more logical.

“Increased Exposure” needs to be explained in the legend or text.

Lines 582-3 – “the fourth from centre annotated text” doesn’t make sense and needs rewording.

Lines 583-6 - “beta-lactamase genes”, “carbapenemase genes” would be sufficient and suggest something like “filled boxes in blue” etc instead of “presence/absence rectangles”.

Line 587 – “minimum inhibitory concentrations of” “-R” is probably not needed here or on the diagram, especially as some isolates are “S” etc. Why is R, S etc shown for these particular antibiotics and the order is not very logical e.g., group antibiotics of the same class. “AMI” is not a standard abbreviation for amikacin.

Fig. 5

The purpose of this figure is not clear and there is overlap with Fig. 4.

The title just needs to say “Antibiotic susceptibility of *E. coli*” and specify if these are all isolates or only those with blaNDM-5 etc.

Fig. 6

Again, there is some overlap with Fig. 4 and see comments on Fig. 4, including the legend.

What do the blue boxes in Clade H show?

Line 596 – this could be deleted – it relates to Methods.

Line 599 – “F or M”

Lines 600-1 – there is no “forth from centre” here - all isolates in the figure are ST167.

7) Supplementary Information

The Supplementary Information is not well thought out or presented, with e.g. tables running across two pages, unnecessary duplication of information, long table and figure titles and legends include information that belongs in Methods. For example: Supplementary Table 3 and Supplementary Table 1 show the same data.

Supplementary Tables 4 and 5 could easily be combined. Gene variants should be distinguished if there are any.

Supplementary Tables 6 – blaNDM-5 could be specified in the title and this column removed.

See also comments above on figures and legends in the main manuscript.

8) Problems with wording/formatting/organization etc.

General points

The nomenclature of blaOXA genes means that it is necessary to specify the variant or family but “blaOXA-48-like” should be followed by “gene” or “genes” every time it is used. “blaOXA-48-like-positive/carrying isolates” (e.g. Lines 140, 142) in particular is problematic. “blaNDM genes”, or specify the blaNDM variant, if known.

ARG names need to be correctly formatted, particularly in the Supplementary material – the (A) of mph(A) and tet(A) should not be in italics, but the rest of the name should be, blaCTX-M-15 has two dashes, bla is in italics but the rest of the name is

not and should be subscript etc.

“positive for” can often be replaced by e.g. “with” or “carrying” (e.g. Line 350).

Specific examples (not exhaustive)

Title – “extended-spectrum beta-lactamase and carbapenemase *E. coli*” does not make sense and a beta symbol should be used.

Line 23 “carriage... in an inpatient SAM centre” doesn’t really make sense.

Line 25 etc– “0-5 years” (Line 45) might be clearer than “0-59 months” but maybe “under 5 years” here.

Line 27 – “by PCR” rather than “via”.

Line 29 – all isolates are from rectal swabs.

Line 32 – it’s really the samples from the children that carry these genes and “and/or” here?

Line 33 – suggest “Many (n=503/729, 69%) children who did not carry bacteria with a carbapenemase gene on admission were found to have such bacteria on discharge”.

Line 42 – “the presence of” is not needed.

Lines 47-67 – this information is not in a very logical order e.g. describe treatment for uncomplicated SAM first, maybe on Line 48, then complicated.

Line 61 - “high rates of carriage of extended-spectrum beta-lactamase genes”

Line 67 – what is meant by “endogenous” here?

Line 68 – suggest “Additionally, health care... with SAM are often”

Line 72 – this study does not address the risk of bacteraemia.

Line 74 – GNB is not defined until Line 340

Line 75 – meaning “dominant clones of *E. coli*”?

Lines 82-87 – “bacteraemia-positive blood cultures” – bacteraemia means bacteria in the blood. If this information has been published previously then it might be better in the Introduction.

Lines 89 etc – use either e.g. “was between 0 and 191 days” or “was 0-191 days”, but here “up to 191 days” is simpler. Also

Line 90 – “hospitalised for 0-5 days, 6-10 days or >11 days”

Line 95 – “Two patients had”.

Line 98 – “and/or” here?

Line 106 – “beta-lactamase genes” is usually used.

Lines 108 – “being attributed to” does not work here.

Line 113 – not clear what is meant by “hospitalisation (patient deterioration)” here. Also Fig. 2.

Line 130 – 673 isolates with a carbapenemase gene?

Line 137 – suggest “Five different *Enterobacter* species with blaNDM were identified”.

“Line 143 – “identification was unknown” does not make sense – meaning “The species could not be identified for...”. Why not?

Line 149 – “within this study” is unnecessary.

Line 155 – the section title “Rectal *E. coli* genomic diversity with blaNDM” needs rewording.

Lines 156-77 – results from on WGS are mixed in with AST results and this could be presented more logically.

“aminoglycoside resistance gene” not “aminoglycoside gene”.

Lines 171-3 repeat some of the same information and could be simplified and condensed.

Line 179- why 183/190 here when Line 62 says that 190 had WGS?

Lines 190, 191 etc – this is more usually written “*E. coli* ST167”.

Line 193-4 – poorly worded – something like “...discharge. This may facilitate transmission into the community as patients return home.” might be more accurate.

Lines 195-6 – clade A contains clade A?

Line 199 – “different patients”?

Line 227 – assigned by whom?

Line 265 – “resistant” is not needed here - MDR means “multidrug resistant”.

Line 329 – “from each child’s caregiver”?

Line 332 – “description...has been described” needs rewording.

Lines 337-9 – there should be a space between numbers and units.

Lines 336-42 – “or vancomycin plus ertapenem”, “respectively” is in the wrong place.

Lines 340-1 – “ESBL producers”, not “ESBLs producers”, also “carbapenemase producers”.

Line 345 – “with TS/72 beads”?

Line 346 – “from samples positive for a blaNDM, blaKPC or blaOXA-48 like gene”.

Line 348 – the species was identified?

Line 351 – “minimum concentrations of 15 antibiotics were”

Lines 356-7 – unclear

Line 374 – what is meant by “cleave” here? Why was this done?

Line 392 – what are “PT”?

Line 407 – “data were”

References – some Refs are missing details e.g. #26 and 27, formatting needs checking (species names etc), change Title Case to Sentence case e.g., #2, #7. etc

Reviewer #2

(Remarks to the Author)

This study was aimed to investigate the carriage of AMR gram-negative pathogen in an inpatient in severe acute malnutrition (SAM) treatment center, particularly comparing rectal colonization of microbiota at admission and discharge. AMR acquisition rate was 69% at SAM treatment center. Among the microbiota, bacteria positive for blaCTX-M-15, blaNDM,

and blaOXA-48-like are prevalent. ST167 E. coli with blaNDM-5 was noted; 11% of children in this study highlighting acquisition of ST167 carrying blaNDM-5 during stay in SAM treatment center.

This is an important observation that health care facilities for children with SAM could cause amplification of multiple resistant bacteria.

Children with severe acute malnutrition are at high risk of mortality and antibiotics prescription is currently recommended even at the risk of increasing antimicrobial resistance in the community. However, this study raised a question whether it can be justified to continue antibiotics prescription which may cause significant clonal expansion of AMR bacteria during hospitalization of patients in SAM treatment center. To answer the question, it is necessary to conduct investigation of clinical burden of antimicrobial prescription to inpatients in SAM treatment center.

Version 1:

Reviewer comments:

Reviewer #1

(Remarks to the Author)

This revised manuscript describes screening of rectal swabs from infants with severe acute malnutrition (SAM), collected on admission, during hospitalization and on discharge from a single inpatient SAM treatment centre in Niger 2016-17. Isolates underwent susceptibility testing and PCR to screen for blaCTX-M-1 group genes and three carbapenemase gene types. Most E. coli with a blaNDM gene underwent short read sequencing, and more isolates have been sequenced using long-read methods since the previous submission. Also, short-read data from all isolates has now been compared with X3 and F type plasmids that were assembled. However, any extra insight provided by this additional information (e.g., are there outbreaks of both strains and plasmids, does this mean that the conclusions might be more specific to this particular hospital), or how the study might be used to improve infection control does not really come out.

Although the manuscript is improved vs. the original version, it still has many problems and more attention to detail and wording is needed to improve accuracy, clarity and readability, as noted below. Line numbers were included in the responses to some (but not all) comments, but these are not always correct, and text quoted in the comments does not always match the text itself.

1) ORIGINAL COMMENT: It is not clear why only these four gene types (Line 342) were chosen for PCR and Results (Line 98) should state which gene types were screened for, not just those that were found.

RESPONSE: The focus for this study was to evaluate and compare the frequency of clinically relevant beta-lactamase ARG. The dominant antibiotic prescribed to patients was ceftriaxone and ESBL production/presence of NDM often renders treatment ineffective. These 4 genes were chosen based on available microbiology and epidemiology data in sub-Saharan Africa. This information has been captured in the first section of results and the results.

ORIGINAL COMMENT: Line 176 mentions that sequencing detected blaCTX-M-55, a variant of blaCTX-M-15 that also has 100% match to both "blaCTX-M-15" primers, and blaCTX-M-27, which would not have been detected by these primers (as it belongs to the blaCTX-M-9 group) and so could be present in other isolates.

RESPONSE: Yes, it is entirely likely that there are multiple different CTX-M variants within the rectal microbiota and the selection of primers also identifies a proportion. We did not have additional information on the 3,0004 rectal swabs beyond the PCR information indicating the sample had bacteria carrying blaCTX-M-1 group. For purified E. coli isolates with available whole genome sequencing data, it was possible to analyse the ARG in more detail, without bias of selection based on primers. We have added to our discussion, that based off supporting WGS data, it is likely that there multiple additional ESBL genes and variants of genes that were not screened for/selected in this study.

ORIGINAL COMMENT: Line 346 - were any samples positive for blaKPC?

RESPONSE: No samples were positive for blaKPC. This is a similar observation to previous studies, particularly where samples originate from patients from sub-Saharan African countries.

NEW COMMENT: I couldn't see any real justification for the choice of ARG types to screen for in the manuscript itself. Something should be added to Results and/or Methods, as well as bringing this up in Discussion. Also, the response to the comment on Line 346 about inclusion of blaKPC seems to contradict the response to the first comment above.

2) Additional Sequencing and plasmid analysis

ORIGINAL COMMENT: Plasmids carrying blaNDM-5 in three isolates where long read data was available were analyzed, but there are also problems with the accuracy of the plasmid sequences and with the analysis. The choice these particular plasmids also appears to be opportunistic and it is not clear that they are particularly representative, as information about plasmid replicons etc in other isolates in the collection is not reported.

ORIGINAL COMMENT: Alignment of the IncFII plasmid sequences from isolates 889 and 2505 provided suggests that most differences between them could be errors (e.g., differences in number of the same base in homopolymer regions, indels in IS with well-defined sequences, some of which would break transposase genes). All differences need to be checked e.g. by mapping raw reads, before sequences are compared to identify only differences that are real.

RESPONSE: We have repeated long-read sequencing with R10 chemistry to reduce sequencing errors (homopolymer errors are particularly associated with R9 chemistry due to the presence of a single reader head for basecalling. R10 introduced a second reader head and a longer barrel to improve basecalling. Due to limited resources, we have performed long-read sequencing on 16 isolates (due to availability & current resources) and this data has been incorporated to the manuscript to replace the old, potentially erroneous data.

NEW COMMENT:

It's not clear how the information gained from this extra long-read sequencing really adds much to the manuscript, without

some further analysis/explanation.

Line 172 says “Long read sequencing of 16 NDM-5 positive isolates”, but Line 349 in Methods says “Long reads of 20 samples”. Which is correct? Did some of the 20 isolates not have blaNDM-5? Were some of them not analysed further? This needs to be explained properly. Were the isolates for which plasmid sequences were provided with the original manuscript (labelled something like p889, p2525, p869) resequenced using R10? They seem to be different from the plasmids shown in Fig. 5. As suggested in original comments, these sequences could be quickly checked for errors by mapping short-reads and included.

It's not clear how the “Representative isolates” (Line 313) were selected for ONT R10. 14/16 are reported as having blaNDM-5 on an X3 plasmid (Lines 173-4), with these plasmids all apparently highly related from the text (although this is not entirely clear). The 2/16 isolates with a fully assembled F-type plasmid carrying blaNDM-5 are both from ST167. Were the available F plasmids and X3 plasmid from ONT R9 data and analysis of short-read data, ST information etc. used to select with isolates went for ONT R10? If so, this should be explained. R10 data may still have some errors.

Lines 169, 170, 183- – “multiple plasmid replicons”. PlasmidFinder can only detect replicon types from short-read data, not plasmids.

Lines 176-7 – two mutations compared with which sequence and what is defined as position 1? In the plncX3-NDM-5a sequence in PV171503, a gene annotated as “ATP-dependent metalloprotease” is at positions 24944-23682, so 25,023 fits with being upstream of this, but a gene annotated as copG is at positions 3008-2724, which does not fit with 38841. What are the original and changed nucleotides? This should be stated. Also, mutations are found “in” not “on” genes (Line 177).

Lines 173-4 – unclear wording - suggest something like “In n=14/16 isolates, blaNDM-5 is found on a X3 plasmid that is predicted to be conjugative using in silico methods”. If the X3 plasmid is related to those carrying blaNDM-5 found globally then there is probably some published information on conjugation of this plasmid type, which could be cited?

Line 178- “the carbapenemase gene” is blaNDM-5? If so, this can just be stated. Again, wording here is awkward, maybe “blaNDM-5 is carried on a 132,316 bp FIA+FIC plasmid that also carries blaCTX-M-15, blaOXA-1 and blaTEM, which is also predicted to be conjugative”. blaCTX-M-15 is the most important clinically and in the context of this manuscript, so should be listed first.

Fig. 5

a) If this is for the n=190 sequence genomes, then why do >190 genomes have IncF? Is this due to multiple IncF replicons in some? This needs to be explained and/or shown in the figure. It is also not possible to tell what combinations of replicons are present from this figure (many seem to have X3+F, Lines 170-2).

b) How do the plasmid sequences shown relate to “plncX3-NDM-5-a” and “plncF-NDM-5” (Line 182 and submitted to GenBank)? Why compare to plasmids without blaNDM-5? It's not clear if start points are set to the same position for each pair of plasmids of the same type. Shading between the X3 and F plasmid to show regions of identity is confusing and not helpful. What % identity is shown by the different shading?

Legend - Line 578 – PlasmidFinder detects replicons. Line 579 - “and” is not needed before groups. A plasmid does not have a “genetic context” – something like “plasmid organisation” should be used.

Line 582 – “172, 507 bp”? The “85 IncF” plasmid also doesn't have blaNDM-5 either?

ORIGINAL COMMENT: Line 445 – BioProject PRJNA1096457 is now available, but no complete assemblies are listed and at least the three complete (and checked) plasmid sequences should be included/submitted separately to GenBank and the accession nos. listed here.

RESPONSE: We apologize for the inconvenience. Genomics data under PRJNA1096457 should now be available, with the supplementary data revised. The plasmid sequences have been uploaded PV171502 (plncX3-NDM5-a) and PV171503 (plncF-NDM5).

NEW COMMENT: PRJNA1096457, which contains 190 sets of Illumina data, PV171502 and PV171503 are now available from GenBank, but at least one example of each distinct plasmid carrying blaNDM-5 generated by ONT should also be submitted to GenBank, preferably whole genome assemblies for all isolates. The PV171502 and PV171503 entries do not state which isolate they were from or have a link to PRJNA1096457.

3) Short-read mapping and Fig. 6

ORIGINAL COMMENT: ...information about plasmid replicons etc in other isolates in the collection is not reported.

RESPONSE: Additionally, we have now mapped the short reads against the two different blaNDM-5 carrying plasmids to determine whether we can detect mapped regions and predict whether the IncX3 or IncF plasmid type is present.

NEW COMMENTS:

I would have thought that the most important thing is to try and see if it is possible to work out which plasmid type/variant carries blaNDM-5 in each isolate and how this relates to the different ST and sample types (A, H, D). From the text (Lines 170-2) it seems that most isolates have an F replicon (as might be expected for E. coli) and many also have an X3 plasmid. From the sequences provided, it seems that blaNDM-5 is in different contexts in the two plasmid types. Is this enough to predict which plasmids is likely to carry blaNDM-5 in each isolate?

Lines 185-6, 186-9 – is this enough to be confident that blaNDM-5 is on the “plncF-NDM-5” type or an X3 plasmid, respectively, in these isolates? If so, something could be said about how many isolates carry the gene on each type. Was short-read data from the the isolates with R10 ONT data mapped? This might also give useful information about what mapping of data from other isolates means.

Lines 186-193 – this need to be with the X3 plasmid information, not after the F plasmid information.

Line 190 – the plasmid sequences don't say anything about whether isolates are heterogenous.

Lines 356-8 – not a complete sentence. You mean one X3 plasmid and one F plasmid, each carrying blaNDM-5, were concatenated. How much sequence do they share around blaNDM-5? What effect would this have?

Line 358 – what is meant by “competitive mapping”?

Figure 6

This might be better as supplementary and presented in landscape format – the isolate numbers on the right are upside down. It is also not clear what order these are in – grouping by ST (and for ST167 etc maybe by clade) and indicating these

ST might show whether particular plasmid type/SNPs are associated with particular ST etc (see information on Lines 164-6, 179-81). It might also be helpful to indicate with isolates were sequenced by ONT and which plasmid carries blaNDM-5 in these. The “Competitive” and “Individual” columns look pretty similar, so it may not be necessary to show both. I think that showing the SNPs for the IncF plasmids with large numbers of SNPs is not useful. a)- f) at the bottom and information on this in the legend are not really needed, as each column has a heading. The legend should instead describe what “mean mapping quality” means and what “complex”, “del”, “ins” and “mnp” mean.

4) “admission swab” “hospitalisation swab” and “patient deterioration”

PREVIOUS COMMENT: Line 113 – not clear what is meant by “hospitalisation (patient deterioration)” here. Also Fig. 2.

RESPONSE: We use the term “hospitalisation” in this study to refer to the time period where the children were admitted to the SAM treatment facility (supported by MSF) for treatment and antibiotic therapy as required. The results section has been rewritten following additional advice from co-authors for clarity & to reduce any redundancy between the text and figures – lines 89-101.

PREVIOUS COMMENT: Line 292 – there is nothing in the Methods or Results about how this deterioration was evaluated.

RESPONSE: We apologise for this oversight. We have added “Rectal swabs were also collected at the time of patient deterioration, which was informed by clinical symptoms guided by clinical discretion” within the Study Design section of the methods. – lines 269-270.

PREVIOUS COMMENT: Line 328 – what is meant by “time of deterioration”?

RESPONSE: Additional samples were collected from patients when they deteriorated. This was decided based upon clinical presentation, clinical signs and symptoms of physical deterioration. If the patient was not improving from medical treatment additional rectal samples (and blood if required) were collected.

NEW COMMENT: This is still confusing – the term “hospitalisation swabs” needs to be better explained (or changed). Line 273 states that all children providing swabs for the study were “admitted” to the hospital (Line 273), which could be taken as meaning that they were all “hospitalised”, but “hospitalisation swabs” were taken from only 209 of the children. On Lines 275-7 something like “Additional rectal swabs (defined as “hospitalisation swabs”) were taken if clinical symptoms suggested that the patient was deteriorating” could be used, if this is correct, and add something to Line 79. Also see Lines 545-6, which might imply that not all patients were admitted.

5) Other scientific points

ORIGINAL COMMENT: Line 66 – what is the definition of multidrug resistant?

RESPONSE: Multidrug resistance is usually defined as bacterial isolates non-susceptible to at least one antibiotic in three or more antibiotic classes to which they were not intrinsically resistant (Magiorakos, 2012). The sentence has been modified by removing beta-lactam to avoid suggesting that only use of one class of antibiotics can facilitate MDR.

NEW COMMENT: MDR is used several times in the revised manuscript, so the explanation and reference should be added. I can't follow the second part of the response, as the original sentence didn't seem to mention beta-lactams.

ORIGINAL COMMENT: Lines 114-20 – did any children with bacteria with a carbapenemase gene on admission have no such bacteria on discharge?

RESPONSE: For 11 children who had with bacteria with a carbapenemase genes on admission were not colonised with bacteria carrying a carbapenemase gene on discharge. This has been added to the first results section, lines 94-96.

NEW COMMENT: The wording of the response does not make sense or vs. the text, which now states that 11/27 children with blaNDM at admission did not have it at discharge, with the numbers being 14/23 for a blaOXA-48-like gene. Given this, the numbers need checking, and the text is repetitive and could be simplified.

ORIGINAL COMMENT: Line 411 – what are these references? Are the sequences available?

RESPONSE: This section has been re-analysed from new sequencing data. All plasmid sequences used for comparative analysis were from NCBI/PLSDB and the accession number listed indicates sequence availability.

NEW COMMENT: Now Lines 337-8. This question was about references for different ST, not plasmids.

6) Remaining problems with wording/formatting/organization etc.

ORIGINAL COMMENT: Title – “extended-spectrum beta-lactamase and carbapenemase E. coli” does not make sense and a beta symbol should be used.

RESPONSE: We have amended the title to: “Acquisition of Extended-Spectrum β -Lactamase and carbapenemase positive E. coli in hospitalised children with severe acute malnutrition in Maradi, Niger”

NEW COMMENT: This could still be improved. Suggest “Acquisition of Escherichia coli carrying extended-spectrum β -lactamase and/or carbapenemase genes by hospitalised children with severe acute malnutrition in Maradi, Niger”

ORIGINAL COMMENT: “blaOXA-48-like-positive/carrying isolates” (e.g. Lines 140, 142) is problematic.

RESPONSE: We have reformatted and edited the text to improve the wording throughout the manuscript.

NEW COMMENT: “blaOXA-48-like-positive/carrying isolates” etc is still used in several places (e.g. Lines 101, 109, 114-5, 117, 119) - this needs to be checked and fixed throughout.

PREVIOUS COMMENT: Line 98 – “and/or” here?

RESPONSE: We thank the reviewer for this comment. This sentence is referring to those children who had a bacterial culture positive for both blaCTX-M-15 and blaOXA-48-like ARGs, as opposed to either or. There was a high carriage rate for blaCTX-M-15 (like) in patients with 76% children having at least one rectal swab positive for bacteria carrying the ESBL (blaCTX-M-15 (like)) ARG.

NEW COMMENT: Now Lines 83-4 “Overall, rectal microbiota from 1,688 samples (1,042 patients, 76%), 338 samples (301 patients, 83 22%), and 339 samples (296 patients, 21.6%) carried clinically relevant blaCTX-M-1-group, blaNDM, and blaOXA-48-like genes, respectively”.

This is still not clear – the “respectively” suggests that the number is for each gene on its own, although “and” instead of “or” is confusing. If 338 is swabs with blaCTX-M-15+blaOXA-48 then this needs to be explained properly. Is 339 this the number for a blaNDM gene alone or blaCTX-M-15+blaNDM? Also, it's probably better to use “carry a blaCTX-M-1 group gene”

rather than “carry blaCTX-M-1 group genes” etc, unless there is evidence for a single isolate having more than one copy? This and use of “and”, “or” and “and/or” needs to be carefully checked throughout the manuscript, to make sure that what is meant is always both correct and clear. e.g. Line 289 – this needs to be “or vancomycin plus ertapenem” not “and”.

ORIGINAL COMMENT: Line 61 - “high rates of carriage of extended-spectrum beta-lactamase genes”

RESPONSE: This has been edited to: “However, high extended-spectrum β -lactamase (ESBL) carriage prevalence may be linked to...” – lines 48-49.

NEW COMMENT “carriage prevalence” is awkward. See suggested wording in original comment.

ORIGINAL COMMENT: Line 95 – “Two patients had”.

RESPONSE: This sentence has been rewritten to “Two patients had only one rectal swab collected upon admission” – lines 79-80.

NEW COMMENT: This is still a bit unclear – all patients should have had only one swab collected on admission? Maybe “For two patients, only a single swab was collected, in both cases at admission”

ORIGINAL COMMENT: Line 106 – “beta-lactamase genes” is usually used.

RESPONSE: The sub-heading has been changed to “Acquisition of Gram-negative bacteria carrying β -lactamase resistance genes” – line 89.

NEW COMMENT: “Beta-lactamase resistance genes” does not make sense – these genes encode a beta-lactamase but give resistance to beta-lactams. “beta-lactamase genes” is simplest.

ORIGINAL COMMENT: Line 155 – the section title “Rectal E. coli genomic diversity with blaNDM” needs rewording.

RESPONSE: The sub-heading has been reworded to: “Genomic diversity of blaNDM carrying E. coli”

NEW COMMENT: “Genomic diversity of E. coli carrying a blaNDM gene” might be clearer.

ORIGINAL COMMENT Line 193-4 – poorly worded – something like “...discharge. This may facilitate transmission into the community as patients return home.” might be more accurate.

RESPONSE: this has been added – lines 154-155.

NEW COMMENT: Lines 155-6? I think that the wording “a high likelihood of community transmission” is too strong – unless you can cite paper(s)/data to support this.

ORIGINAL COMMENT: Lines 195-6 – clade A contains clade A?

RESPONSE: This has been edited for clarity. The tree contains two main branches, of which has been split into clades (sub-clades) A-H on one branch with clade I forming a separate more distant lineage – lines 155-160.

NEW COMMENT: The wording here is still difficult to follow. Suggest something like “...E. coli ST167 population separated by ~450 SNPs (Fig. 4). One branch contains multiple small subclades (A-G) and one large subclade (H, 80 isolates collected December 2016-July 2017; ≤ 20 SNPs) and the other only a single clade (clade I ≤ 20 5 SNPs)”, then discuss blaCTX-M-15 and blaCTX-M-55. Specify which blaTEM gene.

Lines 161-2 – confusing – what is meant by a “strain” here as opposed to the subclades?

ORIGINAL COMMENT: Lines 337-9 – there should be a space between numbers and units.

RESPONSE: This has been corrected throughout.

NEW COMMENT: Still no space in e.g. Lines 174, 177, 179, 350, 356, 579, 581.

ORIGINAL COMMENT: Line 346 – “from samples positive for a blaNDM, blaKPC or blaOXA-48 like gene”.

RESPONSE: This sentence has been changed to: “All phenotypically distinct bacterial colonies were purified from samples positive for a blaNDM, blaKPC and blaOXA-48-like gene for repeat multiplex-PCR” – line 292

NEW COMMENT: No blaKPC genes were detected? Line 289 (not 346) – needs to be “or” or “and/or” here, not “and”.

7) NEW COMMENTS - Discussion

Information is still not well organised and the wording could be improved in places.

Lines 196-8, 199-102 – this seems more like Introduction.

Line 199 – “treatment for complicated SAM”

Line 204 “a globally disseminated”

Line 212 – where is this mentioned in Results?

Line 216 – “ARG carriage”?

Line 219 – bacteria carrying blaCTX-M-15

Line 220 – “an additional concern”

Line 222 – “and suggests” does not make sense here.

Line 224 – “increased carriage of MDR bacteria”

Line 226 – where is this in the Results?

Line 227 – “multiple strains circulating among”

Line 231 – Additionally, 31 children... microbiota, were” Where is this in Results? Or cite Ref 20 earlier.

Line 235 - “received hospitalisation” does not make sense - “are hospitalised”?

Lines 238-9 – wording does not make sense.

Line 248 – “factors such as IPC?” meant here. Has this abbreviation been used before?

Line 251 – first phrase redundant? What “underscores”?

Line 258 – “E. coli carrying blaNDM-5”

Line 259 – what is meant by “mirrored”? specifically blaNDM-5 or different genes?

8) NEW COMMENTS: Methods

Lines 267-283 - order of information could be more logical and reference to complete study details in the previous version seem to have been lost.

Line 274 – make it clear that only rectal swabs were studied here.

Line 277-6 – repeat the same information.

Line 303 – “samples positive for a blaCTX-M-1 group gene, we did not look further at isolate with one of these genes.”?

Line 305 “as described in” is not needed.

Line 310 – which kit?
Line 323 – this not a complete sentence.
Line 331 – “multi-locus sequence types were determined in silico”
Line 332 – “not assigned to a known ST”
Line 334 – “Plasmid replicons were detected”
Line 336 – this is not a complete sentence.
Lines 341-2 – wording does not make sense.

9) NEW COMMENTS: Other minor wording etc

Line 21 – suggest “carriage of Gram-negative bacteria in”
Line 23 – “while in hospital”?
Line 23 - “carrying a blaNDM gene”? Can blaNDM-5 be specified here?
Line 25 – “a blaCTX-M-1 group gene ... a blaNDM gene”? Are all of these known to be blaNDM-5? If so, then “carrying blaNDM-5”.
Line 27 – colonised with what?
Lines 32-4 – is Ref 12 also relevant here?
Lines 47-8 – wording is awkward. If only one of the antibiotics is given, then “and” here should be “or”.
Lines 48-9 – “a high prevalence of carriage of isolates resistant to extended-spectrum beta-lactams (ESBL)” or “with a gene conferring resistance to extended-spectrum beta-lactams (ESBL)”?
Line 50 – “high rates of acquisition”?
Lines 52-3 – “Use of broad-spectrum antibiotics”.
Line 58 - “risk for”?
Line 61 – “carriage of antimicrobial resistant”.
Line 65 – “and/or”?
Line 73 – “the antibiotic most commonly prescribed to this cohort”?
Lines 85-6 – “both a blaCTX-M-1 group gene and a blaNDM gene” etc? Note extra dash in blaCTX-M-1.
Line 92 – “a blaCTX-M-1 group gene”?
Lines 94 – “a blaOXA-48-like gene”. “A blaNDM and/or a blaOXA-48-like gene”?
Line 98 – “blaCTX-M-1-like genes”
Line 105 – “All rectal swabs in which at least one carbapenemase gene was detected”?
Line 109 – you mean that more samples had them?
Line 122 – “E. coli carrying a blaNDM gene”
Line 126 – “patient” not really needed here.
Lines 127-9 – wording can be simplified.
Line 138 – “Levels of resistance in E. coli carrying a blaNDM gene were lowest”
Line 140 – “Rates of resistance to carbapenems were high, as might be expected” Maybe say this first?
Line 143 – “WGS on n=7/21 confirmed”
Line 144 – “a carbapenemase gene”? Which? – both blaOXA-48 like and blaNDM, or just one type?
Lines 144-6 – this would be better in the previous section on E. coli with blaNDM.

10) Figures

Fig. 1

ORIGINAL COMMENT: This is not particularly easy to follow and the information is also in the text. A clearer figure (supplementary?) would mean that most of the detail could be removed from the text.

RESPONSE: We agree that Figure 1 was not particularly easy to follow in the submitted form and there is some duplication with the text in the first results section. The figure has been edited for clarity; however, we still feel this should be Figure 1 in the main text to help set the scene for following dataset.

NEW COMMENT: The wording in most boxes has problems and needs fixing. e.g. “642 children had a rectal sample carrying at least one ARG on admission”. Line 547 “is detailed”. It might be helpful to what was sequenced on this figure, if it can be done simply, as this gets a bit lost in the text.

Fig. 2

Line 552 – can just be “carbapenemase gene” and did each child have only one? Line 555 – fix “others grouped as others”

Line 556 – is this all blaNDM-5 and/or all blaOXA-48?

Fig 3

Three letter abbreviations are more common. e.g. CTX for cefotaxime, as in blaCTX-M, CAZ for ceftazidime.

Fig. 4

A-G called “subclades” in text. Line 570 – reword “blown up... for increased resolution”. Line 572 – fix “hospitalisation”. Line 573 – “of the patient is indicated (F, female; M, male)”. Lines 574-6 – suggest “beta-lactamase genes is” See previous comment on “presence/absence rectangle”, “carbapenemase genes in blue, blaCTX-M genes in green...” etc.

11) Supplementary

The supplementary data section has improved, but still has some problems.

Fig. 1 – its not clear why this has an extra title at the top. “Displaying the” at the start of the legend is not needed and should “genome depth” be “coverage”?

Fig. S2 and Fig. S3 legends could be simplified and condensed e.g. remove text that simply repeats Methods, all isolates are from rectal swabs, discussing admission samples first seems more logical.

Fig. S4 - at this scale it is hard to see anything much on these diagrams and most genes are not labelled. Figure 5 in the main manuscript already shows more detail and more plasmids.

REVIEWER COMMENTS

Reviewer #1 (Remarks to the Author):

This manuscript describes screening of rectal swabs from infants with severe acute malnutrition (SAM), collected on admission, during hospitalization and on discharge from a single inpatient SAM treatment centre in Niger 2016-17. Isolates underwent susceptibility testing and PCR to detect one ESBL gene type and three carbapenemase gene types. Isolates with a carbapenemase gene underwent short read sequencing. Plasmids carrying blaNDM-5 in three isolates where long read data was available were analyzed, but there are also problems with the accuracy of the plasmid sequences and with the analysis. The choice these particular plasmids also appears to be opportunistic and it is not clear that they are particularly representative, as information about plasmid replicons etc in other isolates in the collection is not reported.

Although detecting increasing carriage of carbapenemase genes in bacteria from SAM patient might be important, some of the papers cited already show acquisition of ESBL genes by SAM patients in hospital (e.g. Lines 262-4, Ref 19), and the isolates examined here are now quite old (from 2016-17). The point of including other aspects is not always clear and leads to a lack of coherence.

The manuscript is not well presented, particularly the Supplementary Information, nor well written, and lacks attention to detail/accuracy. Lists of numbers in the text might be better just dealt with in figures, with only the main conclusions stated clearly. The manuscript is also wordy and too long for the amount of data presented. Figures and tables across both the main text and Supplementary are not well thought out and information is unnecessarily duplicated across text, tables and figures.

Main scientific points

1) It is not clear why only these four gene types (Line 342) were chosen for PCR and Results (Line 98) should state which gene types were screened for, not just those that were found. The “blaCTX-M-15” primers (Supplementary Table 7) would also detect other blaCTX-M-1 group genes (maybe also including blaCTX-M-1 itself, though it has one mismatch in the reverse primer) that could not be distinguished without sequencing. This must be explained much more accurately.

The focus for this study was to evaluate and compare the frequency of clinically relevant beta-lactamase ARG. The dominant antibiotic prescribed to patients was ceftriaxone and ESBL production/presence of NDM often renders treatment ineffective. These 4 genes were chosen based on available microbiology and epidemiology data in sub-Saharan Africa. This information has been captured in the first section of results and the results.

We thank the reviewer for pointing out the clarification required from our blaCTX-M primers. Similarly to how we describe blaOXA-48-like genes, we have revised blaCTX-M-15 to blaCTX-M-1-group throughout.

Line 101 – “ARG gene types” would be more accurate, as variants are not defined by PCR. *This has been included to clarify a few modified sentences in the first section of the results.*

Line 176 mentions that sequencing detected blaCTX-M-55, a variant of blaCTX-M-15 that also has 100% match to both “blaCTX-M-15” primers, and blaCTX-M-27, which would not

have been detected by these primers (as it belongs to the blaCTX-M-9 group) and so could be present in other isolates.

Yes, it is entirely likely that there are multiple different CTX-M variants within the rectal microbiota and the selection of primers also identifies a proportion. We did not have additional information on the 3,0004 rectal swabs beyond the PCR information indicating the sample had bacteria carrying blaCTX-M-1 group. For purified E. coli isolates with available whole genome sequencing data, it was possible to analyse the ARG in more detail, without bias of selection based on primers.

We have added to our discussion, that based off supporting WGS data, it is likely that there multiple additional ESBL genes and variants of genes that were not screened for/selected in this study.

2) Sequence analysis (Supplementary Figures 3, 4, 6 and associated text).

The sequence analysis has problems, there are inaccuracies and the text below each of these supplementary figures reads like lab notes, not part of a finished manuscript.

Figs S3 and S4

Alignment of the IncFII plasmid sequences from isolates 889 and 2505 provided suggests that most differences between them could be errors (e.g., differences in number of the same base in homopolymer regions, indels in IS with well-defined sequences, some of which would break transposase genes). All differences need to be checked e.g. by mapping raw reads, before sequences are compared to identify only differences that are real.

Fig. S3

It's not clear what this is showing – it doesn't seem to be the whole plasmid?

The name “pGL2525” is used but not explained.

The ARG gene names are not correctly formatted, differ between the figure and legend and could be written in a more logical order in the legend e.g. the dfrA12-gcuF-aadA2 is a cassette array followed by sul1, Tn3 might actually be Tn2, which is more common, TnAs1 might be Tn21? Tn5403 is mentioned in the legend but not shown in the figure, “ItrA group II intron reverse transcriptase/maturase protein” should just be referred to as “group II Intron” (see Zimmerly Lab for Mobile Group II Introns (ucalgary.ca)),

“tra operon” or “tra gene cluster” would be better than “cassette”.

“dhfr” is incorrect – its dfrA12 as shown in the figure

Which cat gene? – it's probably catB3delta? And it should be blaOXA-1.

IS need to be properly identified by name or number (see ISFinder, <https://www-is.biotoul.fr>)

The region shown is not a “cassette”.

% homology is meaningless – homology is not quantifiable.

The relevance of the results of the comparative analysis is not explained.

Lines 207 – both fully sequenced plasmids are from clade C so maybe it's not surprising that they are probably identical.

Line 209 – “motility” is the wrong word here. “predicted to be conjugative”.

Line 208 – what is meant by FIA/C -FIA-FC? i.e. FIA+FC type replicons? FIA-FII is used elsewhere.

Line 211 - “10 hits contained very similar ...plasmids” does not make sense.

There is no exploration of whether the short reads from the other isolates in Clade C or other clades could be compatible with having the same plasmid e.g. same ARGs sets, mapping etc.

Fig. S4

This map probably corresponds to the plasmids from both 889 and 2525?

The text is hard to read and is not clear why some things are labelled and some not – all of

the ARG shown in Fig. S3 are present as well and the relevance of labelling various RNA is not clear.

The 1.1 kb insert seems to be IS903 (see ISFinder).

Including a quote from a paper is not appropriate – the information needs to be properly related to the data obtained here.

Fig. S6

bleMBL is so named because of its known association with blaNDM and these genes are almost always found together.

The elements upstream of blaNDM-5 are a fragment of ISAba125, well known to be associated with blaNDM genes in many different contexts, derived from Tn125, and here interrupted by ISKpn26. Both of these IS are IS5 family and the extents and names of IS need to be properly identified using ISfinder

The meaning/purpose of including most of the other text here is unclear. This plasmid seems to match a well-known IncX3 plasmid carrying blaNDM-5 (and some other variants), so discussing a comparison with only a few other plasmids is unhelpful. Is 99% identity meant? This should be stated. Using “NDM cassette” to mean the immediate context of blaNDM should be avoided, due to possible confusion with gene cassettes associated with integrons. Again, this plasmid sequence should be carefully checked for errors and corrected if necessary.

We have repeated long-read sequencing with R10 chemistry to reduce sequencing errors (homopolymer errors are particularly associated with R9 chemistry due to the presence of a single reader head for basecalling. R10 introduced a second reader head and a longer barrel to improve basecalling. Due to limited resources, we have performed long-read sequencing on 16 isolates (due to availability & current resources) and this data has been incorporated to the manuscript to replace the old, potentially erroneous data.

All analysis, text, and supplementary information has been replaced with this resubmission, using ISfinder, OriT finder, MOB-suite, Bakta for annotation, and DNaapler for reorientation. The choice of plasmids to provide some contextual analysis was decided upon the highest match (following analysis in PLSDB and mash dist – genomic similarity in public repositories). For our resubmission, we have focused on analysing the available plasmid cohort within this study and report on the presence of genomically similar plasmids in public repositories which suggests global spread of similar plasmids harbouring carbapenemase ARG.

Additionally, we have now mapped the short reads against the two different blaNDM-5 carrying plasmids to determine whether we can detect mapped regions and predict whether the IncX3 or IncF plasmid type is present. We thank the reviewer for suggesting this analysis, and we believe the incorporation of additional data and information on the plasmid content, both from short reads & long reads has improved the quality of our manuscript.

3) Other scientific points

Line 35 – isolates of the same ST would be expected to be “genomically similar”?

We agree, this is redundant and has been deleted.

Line 66 – what is the definition of multidrug resistant?

Multidrug resistance is usually defined as bacterial isolates non-susceptible to at least one antibiotic in three or more antibiotic classes to which they were not intrinsically resistant

(Magiorakos, 2012). The sentence has been modified by removing beta-lactam to avoid suggesting that only use of one class of antibiotics can facilitate MDR.

Line 80 – what type of average? The mean? This should be stated.
This has been changed to mean.

Line 119 – as rectal swabs were taken shouldn't it also be “rectal carriage here?”
This sentence, and the section of results has been modified in many places, “rectal microbiota” has been added for clarification.

Lines 114-20 – did any children with bacteria with a carbapenemase genes on admission have no such bacteria on discharge?
For 11 children who had with bacteria with a carbapenemase genes on admission were not colonised with bacteria carrying a carbapenemase gene on discharge. This has been added to the first results section, lines 94-96.

Line 161 – why is a range given here? This needs to be explained. Is “a few isolates were negative for a bla_NDM gene on repeat PCR” meant here? “few isolates” has a different meaning.
The second batch of MICs was performed at a later time point. Upon re-culturing the E. coli isolates from the storage beads, some were found to be negative on the PCR, indicating the isolate was no longer carrying the bla_NDM gene. Therefore, the MIC was not performed for that antibiotic(s). For clarity, partial datasets have now been removed (this does not change the overall % for data as the difference from the original submission to the cleaned dataset (all at n=217) was very small. Therefore, for clarity, the MIC data for available isolates with a complete set for 15 antibiotics is shown. This is for 217 isolates, please see clarification in the results on lines 138, and amended text in section 137-151.

Line 169-71 – are these 21 of the 248 E. coli with bla_NDM detected by PCR stated at the beginning of this section? Are these 7 isolates part of the 190 that underwent WGS? They all had bla_NDM-5? What could be a possible explanation for the genotype/phenotype discrepancy?
The results for this section is now within a section called “bla_NDM carrying E. coli population”. From the rectal swabs, 248 E. coli positive for bla_NDM via PCR were purified, identified and stored in TS/72 beads. At a later stage, the 248 E. coli isolates were cultured for WGS. The text has been modified for clarity. Likewise, the text in the following section “AMR and genetic mechanisms of resistance in E. coli” has been modified for clarity.

Line 179 – which qnrB gene(s)? Why 183 with WGS here and 190 elsewhere?
This has been added for clarification, and the 183 v 190 corrected. It was 190 with WGS data and this was a previous typo (following repeat sequencing on 7 that had low coverage initially). This section has been edited based on repeat analysis of genome screening using AMRfinderplus. The methods section has been edited and split into sections for clarity. The original short read genomic sequencing and genomic relatedness was performed on a separate occasion (both wet and dry data generation/analysis), and over the past 6-months we have performed some long read sequencing to enable a more robust plasmid analysis and plasmid mapping.

Line 445 – BioProject PRJNA1096457 is now available, but no complete assemblies are listed and at least the three complete (and checked) plasmid sequences should be included/submitted separately to GenBank and the accession nos. listed here.

We apologize for the inconvenience. Genomics data under PRJNA1096457 should now be available, with the supplementary data revised. The plasmid sequences have been uploaded PV171502 (pIncX3-NDM5-a) and PV171503 (pIncF-NDM5).

4) Discussion

This mainly refers to previously published information and does not make it clear how the current study adds to this.

We have amended the discussion section considerably to reduce any redundancy and clarify how our study contributes to the field

Lines 243-4 - Results only cover two ST167 isolates with an F plasmid with blaNDM-5 and one ST1284 isolate with blaNDM-5 on an IncX3 plasmid. No information on plasmids in other isolates is mentioned in Results. How is it known that the IncX3 plasmid is “highly conjugative”?

This section has been rewritten due to the repeat sequencing using Oxford Nanopore’s R10 chemistry. More plasmid analysis is available. Text relating to mobility is predictive only using mobsuite, and this has been clarified in the new results section called “E. coli plasmid population”. We have amended the language away from highly conjugative as we are restricted to in silico prediction data based on the annotated plasmid sequence.

Lines 265-6 – could plasmids be being acquired by resident E. coli?

We thank the reviewer for highlighting this. It is entirely possible that plasmids are being acquired by resident E. coli within the gut/rectal microbiota. We have added a sentence to our discussion to highlight this possibility, lines 221-224.

Lines 268, 273 – ST rather than strains? Different strains can be the same ST. Also “the ST167 global strain”.

We thank the reviewer for raising this discrepancy. The text has been modified accordingly to avoid suggesting all isolates within a ST are the same strain.

Line 272 – what is the evidence for this “dominant circulation” of the IncF plasmid?

We have removed the “carried on a widely disseminated IncF plasmid” part from the sentence due to the limited availability of the plasmid data. The sentence is now:

“However, in addition to the dominant circulation of bla_{NDM-5} in E. coli ST167, we detected several other ST carrying bla_{NDM-5} in different genetic backgrounds, suggesting that there are likely multiple strains circulating amongst immunocompromised children.”

Line 274 – where is this information in Results?

This information has now been incorporated through the additional long-read data available. The list of E. coli STs, all carrying bla_{NDM-5} genes is also stated in the results section “bla_{NDM} carrying E. coli population”

Line 288 – elucidate” is not the best word here.

This sentence has been modified.

Line 292 – there is nothing in the Methods or Results about how this deterioration was evaluated.

We apologise for this oversight. We have added “Rectal swabs were also collected at the time of patient deterioration, which was informed by clinical symptoms guided by clinical discretion” within the Study Design section of the methods. – lines 269-270.

Line 293 – wording problems.

This sentence has been reworded for clarity. “Upon admission and prior to enrolment in the study, informed consent was obtained from each child’s caregiver.” – lines 272-273.

Line 309 – bla_NDM-5?

-5 has been added to: “Our study sheds light on a very serious issue regarding the dissemination of bla_NDM-5 E. coli in Madarounfa District Hospital, Maradi.” – lines 252-253.

5) Methods

Line 322-4 – how were rectal swabs taken? The information on Lines 322 is not needed here, as it is in subsequent sections.

This sentence has been modified accordingly. Rectal swabs were collected according to the standard guidelines whereby the swab was inserted 1-2 inches into the anal canal and gently rotated 3-6 times. The following sentence has been added, as it is not strictly pertinent to add the specific details listed in the above explanation to our manuscript:

“All rectal swabs were collected according to standard anorectal specimen collection guidelines.”

Lines 326-7 – blood samples were not examined as part of this study?

Blood samples were analysed independently from this study. Bloodstream infections were largely community acquired Salmonella species, and this has been published separately.

Line 328 – what is meant by “time of deterioration”?

Additional samples were collected from patients when they deteriorated. This was decided based upon clinical presentation, clinical signs and symptoms of physical deterioration. If the patient was not improving from medical treatment additional rectal samples (and blood if required) were collected.

Line 335 – specify the media.

The media base was Chromatic Detection, and this has been added.

Line 342 – state which was done by PCR, and which were done by multiplex PCR.

This sentence has been separated to indicate which were PCR or multiplex PCR.

Line 346 - were any samples positive for bla_KPC? How many tested positive then negative? Was retesting carried out immediately or after storage?

No samples were positive for bla_KPC. This is a similar observation to previous studies, particularly where samples originate from patients from sub-Saharan African countries. All testing on rectal swabs was performed following culture of the swab after arrival to the laboratory in Cardiff University.

Lines 356-7 – the text in the results implies that E. coli from rectal swabs, not blood cultures, were sequenced. Which is correct?

This was an error, and we apologise for the confusion. Only isolates cultured from rectal swabs were available for sequencing in this study. This has been amended.

Line 368-423 - this section is oddly structured with section headings and overlapping numbered sections. References/website are needed for the various software used.

Following previous comments and suggestions, the results section has been extensively restructured to avoid confusion between sub-headings/sections of the results. All references and websites added for software used.

Line 390 – numbering of bla genes is from protein sequence. ResFinder can be inaccurate as it is based on nucleotide sequence searches.

We agree with this statement. All genomes have been screened for the presence of ARG by AMRfinder plus in addition to ResFinder and we have added data from this to edit our results accordingly.

Line 391 – PlasmidFinder really finds replicon types. The PlasmidFinder results are only presented for the 3 isolates with long read sequencing.

There is now a sub-heading in the results called “E. coli plasmid analysis”. This section has been re-analysed to include PlasmidFinder data from all short reads, and a more detailed analysis of blaNDM-5 plasmids following long-read sequencing using R10 chemistry to reduce incidence of homopolymer error.

Line 411 – what are these references? Are the sequences available?

This section has been re-analysed from new sequencing data. All plasmid sequences used for comparative analysis were from NCBI/PLSDB and the accession number listed indicates sequence availability.

6) Figures

None of the legends are worded well, e.g. there is usually no need to describe the type of figure. See comments above about correctly describing what the primers will actually detect etc.

All figure legends have been edited/changed and now include a small title as required. The methods and results have been amended throughout to clarify the variant/groups that the primers will detect. All figures have been either edited or completely redone.

Fig. 1

This is not particularly easy to follow and the information is also in the text. A clearer figure (supplementary?) would mean that most of the detail could be removed from the text.

We agree that Figure 1 was not particularly easy to follow in the submitted form and there is some duplication with the text in the first results section. The figure has been edited for clarity; however, we still feel this should be Figure 1 in the main text to help set the scene for following dataset. We have made revisions and changed the inclusion of other main figures, see below.

Fig. 2

The meaning of “Deterioration” here is not clear. The legend refers to “Admission, hospitalization (during patient admission) and discharge”, which is not clear either. There is no description of part (b) and the same information seems to be shown in more detail in Fig. 3(b) and (c).

See comments above about correctly describing what the primers will actually detect.

Fig. 3

(a) see comments on Fig. 1

(b) and (c) duplicate Fig. 2b.

Legend – all isolates in the study were “purified from rectal samples”?

Line 152 – “species” not “isolates”?

Figures 2 and 3 have been combined and re-made to reduce overlap/repetition.

Fig. 4

blaNDM-5 does not need to be shown if it is present in all isolates – just state in the title.

The order Admission, Hospitalization, Discharge in the key would be more logical.

“Increased Exposure” needs to be explained in the legend or text.

Lines 582-3 – “the fourth from centre annotated text” doesn’t make sense and needs rewording.

Lines 583-6 - “beta-lactamase genes”, “carbapenemase genes” would be sufficient and suggest something like “filled boxes in blue” etc instead of “presence/absence rectangles”.

Line 587 – “minimum inhibitory concentrations of” “-R” is probably not needed here or on the diagram, especially as some isolates are “S” etc. Why is R, S etc shown for these particular antibiotics and the order is not very logical e.g., group antibiotics of the same class. “AMI” is not a standard abbreviation for amikacin.

Figure 4 has been removed to prioritise the phylogeny for ST167 to avoid overlap of data presentation.

Fig. 5

The purpose of this figure is not clear and there is overlap with Fig. 4.

The title just needs to say “Antibiotic susceptibility of E. coli” and specify if these are all isolates or only those with blaNDM-5 etc.

All isolates purified for analysis contain blaNDM-5 as part of our methodology to select for isolates with a carbapenemase ARG. As the original Figure 4 has been removed, we believe there is no overlap of presented data. The linked source data showing all raw MIC/AST data is also now included.

Fig. 6

Again, there is some overlap with Fig. 4 and see comments on Fig. 4, including the legend. What do the blue boxes in Clade H show?

The blue boxes in Clade H highlight isolates with more genetic distance within the clade, i.e. more pairwise SNPs between these isolates were detected. This has now been summarised in the legend. This is now Figure 4.

Line 596 – this could be deleted – it relates to Methods.

This has been deleted, and a small paraphrased section moved to become the title of the figure.

Line 599 – “F or M”

The legend has been expanded to clarify this. This is now Figure 4

Lines 600-1 – there is no “forth from centre” here - all isolates in the figure are ST167.

This was a typo and has been deleted.

7) Supplementary Information

The Supplementary Information is not well thought out or presented, with e.g. tables running across two pages, unnecessary duplication of information, long table and figure titles and legends include information that belongs in Methods. For example:

Supplementary Table 3 and Supplementary Table 1 show the same data.

Supplementary Tables 4 and 5 could easily be combined. Gene variants should be distinguished if there are any.

Supplementary Tables 6 – bla_{NDM}-5 could be specified in the title and this column removed. See also comments above on figures and legends in the main manuscript.

The supplementary information has been redone, with most data being incorporated into either the methods, or to source data files as recommended by the journal.

8) Problems with wording/formatting/organization etc.

General points

The nomenclature of bla_{OXA} genes means that it is necessary to specify the variant or family but “bla_{OXA}-48-like” should be followed by “gene” or “genes” every time it is used.

“bla_{OXA}-48-like-positive/carrying isolates” (e.g. Lines 140, 142) in particular is problematic. “bla_{NDM} genes”, or specify the bla_{NDM} variant, if known.

ARG names need to be correctly formatted, particularly in the Supplementary material – the (A) of mph(A) and tet(A) should not be in italics, but the rest of the name should be, bla_{CTX}-M-15 has two dashes, bla is in italics but the rest of the name is not and should be subscript etc.

“positive for” can often be replaced by e.g. “with” or “carrying” (e.g. Line 350).

Specific examples (not exhaustive)

We have reformatted and edited the text to improve the wording throughout the manuscript.

The bla_{NDM} gene variant is always specified, if known.

Title – “extended-spectrum beta-lactamase and carbapenemase E. coli” does not make sense and a beta symbol should be used.

We have amended the title to:

“Acquisition of Extended-Spectrum β -Lactamase and carbapenemase positive E. coli in hospitalised children with severe acute malnutrition in Maradi, Niger”

Line 23 “carriage... in an inpatient SAM centre” doesn’t really make sense.

This has been edited and rewritten.

Line 25 etc– “0-5 years” (Line 45) might be clearer than “0-59 months” but maybe “under 5 years” here.

This sentence has been rewritten and the abstract heavily reduced as per journal guidelines to fit within 150 words.

Line 27 – “by PCR” rather than “via”.

This sentence has been rewritten and the abstract heavily reduced as per journal guidelines to fit within 150 words.

Line 29 – all isolates are from rectal swabs.

Yes, all isolates are rectal swab. This has been edited with the word rectal removed to: “E. coli isolates carrying bla_{NDM} genes were selected for whole genome sequencing...”

Line 32 – it’s really the samples from the children that carry these genes and “and/or” here?

We apologise for this oversight in the language. This has been corrected and edited to: “...1,042, 338 and 339 children harbouring bacteria positive for bla_{CTXM-15}, bla_{NDM}, and bla_{OXA-48}-like genes respectively.”

Line 33 – suggest “Many (n=503/729, 69%) children who did not carry bacteria with a carbapenemase gene on admission were found to have such bacteria on discharge”.

Thank you. This suggestion has been incorporated to the text, see lines 34-35.

Line 42 – “the presence of” is not needed.

This has been deleted.

Lines 47-67 – this information is not in a very logical order e.g. describe treatment for uncomplicated SAM first, maybe on Line 48, then complicated.

This has been restructured so the sentence describing treatment for uncomplicated SAM is before complicated.

Line 61 - “high rates of carriage of extended-spectrum beta-lactamase genes”

This has been edited to:

“However, high extended-spectrum β -lactamase (ESBL) carriage prevalence may be linked to...” – lines 48-49.

Line 67 – what is meant by “endogenous” here?

In this context, endogenous has been used to indicate that it is possible that transmission events/microbial transfer/translocation may be occurring within the host For clarity, we have since amended this sentence to:

“Broad spectrum β -lactam antibiotic usage may select for intestinal carriage of multidrug resistant (MDR) gut bacteria, predominantly E. coli and K. pneumoniae, which may increase the likelihood of transmission events within and beyond the gut microbial community^{18,19}.” - lines 52-55.

Line 68 – suggest “Additionally, health care... with SAM are often”

Thank you for this suggestion. We have amended the sentence to:

“Furthermore, healthcare facilities for children presenting with SAM are often overcrowded, with limited infrastructure and resources for adequate infection prevention and control practices” – lines 56-58

Line 72 – this study does not address the risk of bacteraemia.

Yes, apologies for this oversight and incorrect link to bacteraemia. This has been edited to: There is limited data on antimicrobial-resistant bacterial carriage in children presenting with SAM in Niger” – lines 60-61.

Line 74 – GNB is not defined until Line 340

We apologise for this oversight. The first introduction of GNB appears on line 64, and the full use has been added here, with further removal of the full use on line 330.

Line 75 – meaning “dominant clones of E. coli”?

Thank you for raising this query. We have rephrased the latter half of the sentence for clarity: “...and evaluated dominant clones of E. coli with respect to AMR carriage”.

Lines 82-87 – “bacteraemia-positive blood cultures” – bacteraemia means bacteria in the blood. If this information has been published previously then it might be better in the Introduction.

This sentence has been modified to remove “positive” and moved to the end of the introduction where we introduce the study – lines 61-63.

Lines 89 etc – use either e.g. “was between 0 and 191 days” or “was 0-191 days”, but here “up to 191 days” is simpler. Also Line 90 – “hospitalised for 0-5 days, 6-10 days or >11 days”

Thank you for suggesting these alterations. The text has been modified in both sentences as suggested – lines 75-76.

Line 95 – “Two patients had”.

This sentence has been rewritten to “Two patients had only one rectal swab collected upon admission” – lines 79-80.

Line 98 – “and/or” here?

*We thank the reviewer for this comment. This sentence is referring to those children who had a bacterial culture positive for both *bla*_{CTX-M-15} and *bla*_{OXA-48-like} ARGs, as opposed to either or. There was a high carriage rate for *bla*_{CTX-M-15} (like) in patients with 76% children having at least one rectal swab positive for bacteria carrying the ESBL (*bla*_{CTX-M-15} (like)) ARG.*

Line 106 – “beta-lactamase genes” is usually used.

Thank you. The sub-heading has been changed to “Acquisition of Gram-negative bacteria carrying β -lactamase resistance genes” – line 89.

Lines 108 – “being attributed to” does not work here.

*This has been changed to “with 97% (n=621/642) carrying *bla*_{CTX-M-1} -group” – line 91.*

Line 113 – not clear what is meant by “hospitalisation (patient deterioration)” here. Also Fig. 2.

We use the term “hospitalisation” in this study to refer to the time period where the children were admitted to the SAM treatment facility (supported by MSF) for treatment and antibiotic

therapy as required. The results section has been rewritten following additional advice from co-authors for clarity & to reduce any redundancy between the text and figures – lines 89-101.

Line 130 – 673 isolates with a carbapenamase gene?

Yes, this has been added to the sentence: “In total, 673 bacterial isolates carrying a carbapenamase ARG were recovered, encompassing 15 different species across eight genera” – lines 105-107.

Line 137 – suggest “Five different Enterobacter species with bla_{NDM} were identified”.

Thank you for this suggestion. This sentence on lines 134-145 has been edited to: “Five different Enterobacter species with a bla_{NDM} ARG variant, along with Citrobacter, Pantoea, and Pseudomonas species (Figure 3b) were identified.” – line 111.

“Line 143 – “identification was unknown” does not make sense – meaning “The species could not be identified for...”. Why not?

This sentence has been edited to: “For 59/731 bacterial isolates, (n=35 carrying bla_{NDM} and n=24 carrying bla_{OXA-48}-like), it was not possible to identify the bacterial species using MALDI-TOF MS from three independent tests.”

For some bacterial species, it is possible they were/are not in the Bruker MALDI database, resulting in a final reported identification of “no peaks”. These isolates were not characterised further in this study as E. coli were the species of interest.

Line 149 – “within this study” is unnecessary.

This has been deleted.

Line 155 – the section title “Rectal E. coli genomic diversity with bla_{NDM}” needs rewording.

Yes, thank you for identifying this. The sub-heading has been reworded to: “Genomic diversity of bla_{NDM} carrying E. coli”

Lines 156-77 – results from on WGS are mixed in with AST results and this could be presented more logically. “aminoglycoside resistance gene” not “aminoglycoside gene”. *The results sub-headings and structure has been modified. We have a section now called “AMR and genetic mechanisms of resistance in E. coli”. This section has been edited to improve clarity.*

Lines 171-3 repeat some of the same information and could be simplified and condensed.

This section has been edited to review and condense.

Line 179- why 183/190 here when Line 62 says that 190 had WGS?

This sentence has been removed during the restructure, and throughout the text, the numbers of E. coli isolates with WGS and MIC data has been made clear.

Lines 190, 191 etc – this is more usually written “E. coli ST167”.

This has been corrected throughout as appropriate.

Line 193-4 – poorly worded – something like “...discharge. This may facilitate transmission into the community as patients return home.” might be more accurate.

Thank you for this suggestion, this has been added – lines 154-155.

Lines 195-6 – clade A contains clade A?

This has been edited for clarity. The tree contains two main branches, of which has been split into clades (sub-clades) A-H on one branch with clade I forming a separate more distant lineage – lines 155-160.

Line 199 – “different patients”?

This has been added to clarify – line 160.

Line 227 – assigned by whom?

This section of the manuscript has been edited and rewritten. All ST assignment was performed by Enterobase following upload of reads for processing. This has also been clarified in the methods – line 325.

Line 265 – “resistant” is not needed here - MDR means “multidrug resistant”.

The word resistant after MDR has been removed – line 217.

Line 329 – “from each child’s caregiver”?

This has been changed to “informed consent was obtained from parents/primary carer – line 272

Line 332 – “description...has been described” needs rewording.

This section has also been edited following recommendation from the journal, providing more details on the ethical bodies and study identifiers – lines 273-276.

Lines 337-9 – there should be a space between numbers and units.

This has been corrected throughout.

Lines 336-42 – “or vancomycin plus ertapenem”, “respectively” is in the wrong place.

The word respectively has been removed from the parentheses in the sentence – line 282

Lines 340-1 – “ESBL producers”, not “ESBLs producers”, also “carbapenemase producers”.

This has been corrected – line 284

Line 345 – “with TS/72 beads”?

This has been changed to “with TS/72 beads” – line 291

Line 346 – “from samples positive for a bla_{NDM}, bla_{KPC} or bla_{OXA-48} like gene”.

This sentence has been changed to: “All phenotypically distinct bacterial colonies were purified from samples positive for a bla_{NDM}, bla_{KPC} and bla_{OXA-48}-like gene for repeat multiplex-PCR” – line 292

Line 348 – the species was identified?

This sentence has been modified to begin with “The bacterial species for isolates positive...were identified...” – line 294

Line 351 – “minimum concentrations of 15 antibiotics were”

This has been amended “The minimum inhibitory concentration of 15 antibiotics were determined” – line 297

Lines 356-7 – unclear
This sentence has been deleted.

Line 374 – what is meant by “cleave” here? Why was this done?
NCBI does not allow submissions with contigs <200 bp and this is a post-processing step of the draft genome assemblies. This sentence has been edited for clarity and the use of cleave replaced with remove.

Line 392 – what are “PT”?
This has been replaced with plasmid incompatibility types

Line 407 – “data were”
This has been changed from was to were.

References – some Refs are missing details e.g. #26 and 27, formatting needs checking (species names etc), change Title Case to Sentence case e.g., #2, #7. etc
References have been edited. More references supporting all the bioinformatics tools and databases used have been added.

Reviewer #2 (Remarks to the Author):

This study was aimed to investigate the carriage of AMR gram-negative pathogen in an inpatient in severe acute malnutrition (SAM) treatment center, particularly comparing rectal colonization of microbiota at admission and discharge. AMR acquisition rate was 69% at SAM treatment center. Among the microbiota, bacteria positive for blaCTX-M-15, blaNDM, and blaOXA-48-like are prevalent. ST167 E. coli with blaNDM-5 was noted; 11% of children in this study highlighting acquisition of ST167 carrying blaNDM-5 during stay in SAM treatment center.

This is an important observation that health care facilities for children with SAM could cause amplification of multiple resistant bacteria.

Children with severe acute malnutrition are at high risk of mortality and antibiotics prescription is currently recommended even at the risk of increasing antimicrobial resistance in the community. However, this study raised a question whether it can be justified to continue antibiotics prescription which may cause significant clonal expansion of AMR bacteria during hospitalization of patients in SAM treatment center. To answer the question, it is necessary to conduct investigation of clinical burden of antimicrobial prescription to inpatients in SAM treatment center.

We thank reviewer 2 for their summary. We believe that modifications to the manuscript as suggested by reviewer 1 fully cover topics within this summary. We agree with the final point raised by reviewer 2 regarding the need for future work to clinically investigate the clinical burden of antimicrobial prescription to inpatients in SAM treatment centres. Further assessments of access to diagnostics for improved antimicrobial stewardship are also warranted. We believe our observational data will prompt funding support in this area.

REVIEWER COMMENTS

Reviewer #1 (Remarks to the Author):

This revised manuscript describes screening of rectal swabs from infants with severe acute malnutrition (SAM), collected on admission, during hospitalization and on discharge from a single inpatient SAM treatment centre in Niger 2016-17. Isolates underwent susceptibility testing and PCR to screen for blaCTX-M-1 group genes and three carbapenemase gene types. Most E. coli with a blaNDM gene underwent short read sequencing, and more isolates have been sequenced using long-read methods since the previous submission. Also, short-read data from all isolates has now been compared with X3 and F type plasmids that were assembled. However, any extra insight provided by this additional information (e.g., are there outbreaks of both strains and plasmids, does this mean that the conclusions might be more specific to this particular hospital), or how the study might be used to improve infection control does not really come out.

Although the manuscript is improved vs. the original version, it still has many problems and more attention to detail and wording is needed to improve accuracy, clarity and readability, as noted below. Line numbers were included in the responses to some (but not all) comments, but these are not always correct, and text quoted in the comments does not always match the text itself.

We have amended our manuscript accordingly in this second revision according to the additional comments & any new comments arising from our revised submission. Line numbers with tracked changes and clean v full versions are difficult to follow, perhaps depending on which document is being viewed, and the transition from Word to PDF. This was also evident our end during revision rounds. In addition to line numbers, a description is included in this response to help track the change and position in the article.

We have been through our manuscript carefully to tighten up the language for consistency and accuracy throughout. We appreciate your comment in relation to how this study might be used to improve infection control. It is important to us to improve this in our study. Although we have included this in our discussion section, we have amended our conclusion to emphasise this, see lines 272=288 and section below:

“Conclusion

Access to safe water, sanitation, and hygiene (WASH) is vital to prevent the spread of AMR bacteria. As IPC is intrinsically linked to WASH factors, this study underscores the urgent need to strengthen IPC measures in LMIC treatment facilities. Evidence-based IPC programs should be reviewed and prioritized in inpatient settings, particularly for immunocompromised populations such as children presenting with complicated SAM. Determining whether correlations exist between bacterial acquisition and antibiotic therapy, medical procedures, and hygiene practices could guide treatment guidelines and policymakers in reconsidering antibiotic treatment protocols for complicated SAM to mitigate the spread of AMR...”

1) ORIGINAL COMMENT: It is not clear why only these four gene types (Line 342) were chosen for PCR and Results (Line 98) should state which gene types were screened for, not just those that were found.

RESPONSE: The focus for this study was to evaluate and compare the frequency of clinically relevant beta-lactamase ARG. The dominant antibiotic prescribed to patients was ceftriaxone and ESBL production/presence of NDM often renders treatment ineffective. These 4 genes were chosen based on available microbiology and epidemiology data in sub-Saharan Africa. This information has been captured in the first section of results and the results.

ORIGINAL COMMENT: Line 176 mentions that sequencing detected blaCTX-M-55, a variant of blaCTX-M-15 that also has 100% match to both “blaCTX-M-15” primers, and blaCTX-M-27, which would not have been detected by these primers (as it belongs to the blaCTX-M-9 group) and so could be present in other isolates.

RESPONSE: Yes, it is entirely likely that there are multiple different CTX-M variants within the rectal microbiota and the selection of primers also identifies a proportion. We did not have additional information on the 3,0004 rectal swabs beyond the PCR information indicating the sample had bacteria carrying blaCTX-M-1 group. For purified E. coli isolates with available whole genome sequencing data, it was possible to analyse the ARG in more detail, without bias of selection based on primers. We have added to our discussion, that based off supporting WGS data, it is likely that there multiple additional ESBL genes and variants of genes that were not screened for/selected in this study.

ORIGINAL COMMENT: Line 346 - were any samples positive for blaKPC?

RESPONSE: No samples were positive for blaKPC. This is a similar observation to previous studies, particularly where samples originate from patients from sub-Saharan African countries.

NEW COMMENT: I couldn't see any real justification for the choice of ARG types to screen for in the manuscript itself. Something should be added to Results and/or Methods, as well as bringing this up in Discussion. Also, the response to the comment on Line 346 about inclusion of blaKPC seems to contradict the response to the first comment above.

NEW RESPONSE: We have now reinforced our justification for the selection as CTX-M-1 group in the discussion, as this fits nicely into the section.

We have also expanded our explanation in the methods including the addition of supporting references. Please see lines 317-320:

“The *bla*_{CTX-M-1} group was selected as the target ESBL to screen for in this study due to reported high prevalence and global dissemination³⁵. *bla*_{NDM} and *bla*_{OXA-48}-like ARG were screened for as they are often reported as the most prevalent carbapenemase genes in Africa^{37,38}. Although *bla*_{KPC} carbapenemase gene is not endemic to Africa³⁸, it was also screened for in this study as part of a multiplex PCR.”

To further clarify, *bla*_{KPC} was included in our study for ARG screening rectal samples for two reasons; 1 - The prevalence of *bla*_{KPC} is reported to be low in sub-Saharan African in the literature, but evidence is very limited across many countries, with little in the literature screening rectal/carriage samples for *bla*_{KPC}. 2 – *bla*_{KPC} is within a multiplex PCR enabling detection of *bla*_{NDM}, *bla*_{OXA-48}-like *bla*_{KPC} variants simultaneously from the same reaction. *bla*_{KPC} was therefore included and our 0% prevalence finding was reported as a finding.

2) Additional Sequencing and plasmid analysis

ORIGINAL COMMENT: Plasmids carrying blaNDM-5 in three isolates where long read data was available were analyzed, but there are also problems with the accuracy of the plasmid sequences and with the analysis. The choice these particular plasmids also appears to be opportunistic and it is not clear that they are particularly representative, as information

about plasmid replicons etc in other isolates in the collection is not reported.

ORIGINAL COMMENT: Alignment of the IncFII plasmid sequences from isolates 889 and 2505 provided suggests that most differences between them could be errors (e.g., differences in number of the same base in homopolymer regions, indels in IS with well-defined sequences, some of which would break transposase genes). All differences need to be checked e.g. by mapping raw reads, before sequences are compared to identify only differences that are real.

RESPONSE: We have repeated long-read sequencing with R10 chemistry to reduce sequencing errors (homopolymer errors are particularly associated with R9 chemistry due to the presence of a single reader head for basecalling. R10 introduced a second reader head and a longer barrel to improve basecalling. Due to limited resources, we have performed long-read sequencing on 16 isolates (due to availability & current resources) and this data has been incorporated to the manuscript to replace the old, potentially erroneous data.

NEW COMMENT:

It's not clear how the information gained from this extra long-read sequencing really adds much to the manuscript, without some further analysis/explanation.

Line 172 says "Long read sequencing of 16 NDM-5 positive isolates", but Line 349 in Methods says "Long reads of 20 samples". Which is correct? Did some of the 20 isolates not have blaNDM-5? Were some of them not analysed further? This needs to be explained properly. Were the isolates for which plasmid sequences were provided with the original manuscript (labelled something like p889, p2525, p869) resequenced using R10? They seem to be different from the plasmids shown in Fig. 5. As suggested in original comments, these sequences could be quickly checked for errors by mapping short-reads and included.

NEW RESPONSE: We apologise for the confusion. We attempted re-sequencing on 20 isolates; however, 3 genomes were insufficient coverage to assemble the expected genome size (+/-10% and we excluded them from any analysis). The fourth genome we excluded from re-analysis was isolate 285, however the assembly indicated that the NDM-5 plasmid was no longer present, limiting our analysis, therefore 16 plasmid sequences from 16 *E. coli* isolates passed QC metrics for additional plasmid analysis in this study. This has been amended in our methods.

Source Data 6 lists all sequencing data available, including original sequencing datasets and the additional 16 WGS.

Isolate 869 was available for re-sequencing and was included. We have now performed a comparative analysis to the original plasmid sequence and the repeat plasmid for this isolate. Unfortunately, isolate growth, DNA extraction and degree of success of library preparation/sequencing for 2525 and 2623 was very limited – different freezer location & movement of archived project due to sampling chronology (i.e. collected at a later time period) - cell culture/viability was poor. We attempted WGS (for 2525 and 2623 used for variant calling major ST groups) however there was not sufficient data for analysis or assembly. We therefore decided to expand our selection of available isolates within the collection to maximise our coverage of the different STs/strains (using the phylogenies generated) across the enrolment period. This resulted in n=16 isolates for WGS, which was also an achievable number with limited resources available for this additional sequencing.

They seem to be different from the plasmids shown in Fig. 5. As suggested in original comments, these sequences could be quickly checked for errors by mapping short-reads and included.

NEW RESPONSE: These plasmids are different to the original submission containing R9 and Illumina data. To avoid including any data or extended plasmid analysis with erroneous sequences (plasmid assemblies with homopolymer sequences present etc), we opted to re-sequence a selection and start the analysis a fresh (see extended explanation for this and the selection process above).

We have however checked these plasmids for errors to see if we can include them/what type of plasmids with NDM they are. We have compared the original R9+ Illumina sequences to our repeat R10 using the latest chemistry sequences. We can conclude that because one of the R9 containing plasmid sequences contained some additional and rearranged sequences in the IncX3 plasmid (against the consensus of all other R9 and R10 sequences which assembled a 46,161 bp plasmid) we have continued to focus our plasmid analysis from our repeat sequencing data. We have however included the data into our data availability for all sequence data generated during this study and is available. When directly comparing the plasmid from one isolate sequencing on all three occasions (Illumina short read, R9 ONT and R10 ONT), we have an extremely similar plasmid (within 4 bp), suggesting the original R9 + Illumina assemblies were not too erroneous.

Nevertheless, our repeat sequencing (Fig 5b and c) was selected for all plasmid analysis. To avoid a mismatch of technologies, as there can be differences between chemistries and approaches on *de novo* assemblies – which are a hypothetical attempt to reconstruct the genome, we performed mapping to two different plasmid types using plasmids assembled from the same sequencing approach = R10. In the case of IncX3 where there was n= 14 different plasmids assembled – there was a large consensus in the assembly plasmid size and structure, therefore we proceeded to utilise this data for our plasmid analyses.

It's not clear how the “Representative isolates” (Line 313) were selected for ONT R10. 14/16 are reported as having blaNDM-5 on an X3 plasmid (Lines 173-4), with these plasmids all apparently highly related from the text (although this is not entirely clear). The 2/16 isolates with a fully assembled F-type plasmid carrying blaNDM-5 are both from ST167. Were the available F plasmids and X3 plasmid from ONT R9 data and analysis of short-read data, ST information etc. used to select with isolates went for ONT R10? If so, this should be explained. R10 data may still have some errors.

NEW RESPONSE: Our explanation above provides further explanation as to how the isolates were selected for ONT R10. From the 16 isolates, there were 10 ST167 isolates, n=8/10 ST167 had a 46,161bp IncX3 plasmid with blaNDM-5, whereas n=2/10 ST167 had a 132,316bp IncF plasmid with blaNDM-5. Of the 10 ST167 isolates, isolates were selected across the phylogeny (from clades A, B, F, G, H and I). Three of the 16 for R10 ONT were from the ST group ST1284, all contained blaNDM-5 on a 46,161bp plasmid. One of the 16 R10 ONT was a ST2083 with a blaNDM-5 gene on a 46,161bp plasmid. The two final isolates from the 16 selected for R10 ONT were ST11025 and contained the blaNDM-5 plasmid on a 46,161bp plasmid. This was summarised in the results however we appreciate it was clear and has been modified to – “The majority (n=14/16) were found on *in silico* predicted (*oriT*, MOBP relaxase) conjugative IncX3 plasmids (46,161bp), and present across four different ST groups with ONT data available enabling plasmid assembly (*E. coli* ST167, n=10; ST1284, n=3, ST11025, n=2, and ST2083, n=1; Fig. 5b, Supplementary Figure S4).”

Further, the methods section has been modified accordingly, and Source Data 6 summarises all relevant information: “Long reads of 16 samples (n=10 ST167, n=3 ST1284, n=2 ST11025, n=1 ST2083; Source Data 6) were assembled using Flye (v2.9.4-b1799) and

polished with medaka (v1.12.1) after removing poor quality reads (Q12/Q15 and 1000bp length) with nanoq (v0.10.0).”

Lines 169, 170, 183- – “multiple plasmid replicons”. PlasmidFinder can only detect replicon types from short-read data, not plasmids.

NEW RESPONSE: This has been corrected as suggested. Where long read data enabled plasmid assembly and detailed plasmid analysis, we have retained the use of plasmids. All other cases, where PlasmidFinder results were used (i.e. for the larger n=190 Illumina dataset) “plasmid replicons” has been used – “The *E. coli* genomes contained multiple plasmid replicons, with IncF, Col and IncX3 being the most common detected via screening short read genomes (Fig. 5a). IncF plasmid replicons were detected in 178 genomes, while IncX3 plasmid replicons were detected in 118 genomes”.

Lines 176-7 – two mutations compared with which sequence and what is defined as position 1? In the pIncX3-NDM-5a sequence in PV171503, a gene annotated as “ATP-dependent metalloprotease” is at positions 24944-23682, so 25,023 fits with being upstream of this, but a gene annotated as copG is at positions 3008-2724, which does not fit with 38841. What are the original and changed nucleotides? This should be stated.

NEW RESPONSE: We have checked the annotated files and for IncX3 plasmids, it appears there are two copG regions, however there is discrepancy in the available annotation databases. The first is at position 3008-2724 as identified and at position 38,715 to 38,951 depending on the nucleotide certain sequences will flag the gene to be either copG or rafH. This is also now reflected in the text and figure 5 for clarity: “On the IncX3 plasmids (n = 14 *E. coli* isolates with IncX3 plasmids assembled), two mutations were observed upstream of ATP-dependent metalloprotease gene (25,023 bp) and at position 38,841 bp in the CopG or rafH gene (depending on the nucleotide in position 38,841 bp; our reference plasmid used for mapping annotated as CopG with nucleotide A in position 38,841 whereas the mutation/change to position G caused a change from asparagine to glycine. For those isolates A – G the gene annotated as an RfaH, which when exploring the annotation further, this could be a fused gene, or perhaps more likely a chimeric gene (a result of genomic rearrangement) phenomenon causing an overlap. For clarity in the text, we have annotated the reference plasmid used for mapping as a transcriptional regulator (seen in the main text and figure 5b) – an online search on uniprot and other databases have indicated that this gene is part of NusG and others have labelled this as a hypothetical protein. Regardless, we have also now specified the nucleotide change for context.

Also, mutations are found “in” not “on” genes (Line 177).

NEW RESPONSE: This has been corrected. – “and in the CopG gene (38,841bp).”

Lines 173-4 – unclear wording - suggest something like “In n=14/16 isolates, bla_{NDM-5} is found on a X3 plasmid that is predicted to be conjugative using *in silico* methods”.

NEW RESPONSE: We agree, and we have amended the sentence to – “In n=14/16 isolates (of different ST groups with available ONT data; *E. coli* ST167, n=10; ST1284, n=3, ST11025, n=2, and ST2083, n=1”). bla_{NDM-5} was found on a IncX3 plasmid (46,161bp) that is predicted to be conjugative (*oriT*, MOB_P relaxase) using *in silico* methods”.

If the X3 plasmid is related to those carrying bla_{NDM-5} found globally then there is probably some published information on conjugation of this plasmid type, which could be cited?

NEW RESPONSE: We also agree there is literature evidencing and reviewing the high conjugative ability of IncX3 plasmids. Our discussion includes a (slightly amended for this revision) sentence relating to this too – “Our study found a high rate of rectal acquisition of the globally prevalent high-risk *E. coli* strain ST167 carrying bla_{NDM-5} which was largely located on a globally disseminated and often highly conjugative IncX3 plasmid^{25,26}.”

Line 178- “the carbapenemase gene” is bla_{NDM-5}? If so, this can just be stated.

NEW RESPONSE: This has been amended to– “Two *E. coli* ST167 isolates carried bla_{NDM-5} on a 132,316 bp IncFIA/FIC plasmid”.

Again, wording here is awkward, maybe “blaNDM-5 is carried on a 132,316 bp FIA+FIC plasmid that also carries blaCTX-M-15, blaOXA-1 and blaTEM, which is also predicted to be conjugative”. blaCTX-M-15 is the most important clinically and in the context of this manuscript, so should be listed first.

NEW RESPONSE: We have amended the text in this sentence as suggested to – “Two *E. coli* ST167 isolates carried blaNDM-5 on a 132,316 bp IncFIA/FIC plasmid that also carries blaCTX-M-15, blaOXA-1 and blaTEM, which is also predicted to be conjugative (*oriT*, MOB relaxase; Fig. 5b, Fig. 6a-c, Supplementary Figure S4”,

Fig. 5

a) If this is for the n=190 sequence genomes, then why do >190 genomes have IncF? Is this due to multiple IncF replicons in some? This needs to be explained and/or shown in the figure. It is also not possible to tell what combinations of replicons are present from this figure (many seem to have X3+F, Lines 170-2).

NEW RESPONSE: Thank you for raising this query. It does suggest we need to amend our figure to incorporate this data more clearly. Multiple *E. coli* isolates contain >1 IncF replicon detected. Yes, several (n = X) isolates have IncX3 and IncF replicon types, although as indicated by our mapping (Fig. 6), the IncF present in isolates with the IncX3 plasmids do not map reliably to the IncF containing a blaNDM-5 gene suggesting they are different IncF plasmids. We have amended Figure 5 to incorporate this data more clearly.

b) How do the plasmid sequences shown relate to “pIncX3-NDM-5-a” and “pIncF-NDM-5” (Line 182 and submitted to GenBank)? Why compare to plasmids without blaNDM-5? It’s not clear if start points are set to the same position for each pair of plasmids of the same type. Shading between the X3 and F plasmid to show regions of identity is confusing and not helpful. What % identity is shown by the different shading?

NEW RESPONSE: We have remade figure 5 to address other comments raised and provide clarity. We have removed a comparison to plasmids without NDM-5 and have prioritised summarising the plasmid content in Figure 5 and, providing an annotation/description of the major distinct plasmid types identified in this cohort with NDM-5 (by long read sequencing). Our source data now provides sufficient links between all sequence data types and isolates within the collection.

Legend - Line 578 – PlasmidFinder detects replicons. Line 579 - “and” is not needed before groups. A plasmid does not have a “genetic context” – something like “plasmid organisation” should be used.

NEW RESPONSE: The legend has been changed to – “Fig. 5 *E. coli* plasmid types. a) plasmid replicons detected across the n=190 genomes grouped to eight categories b) plasmid organisation of the 46,161bp IncX3 carrying blaNDM-5 plasmid (isolate 342) alongside an IncX3 plasmid sequenced from isolate 285 which was negative for blaNDM-5 upon reculture. The 132,316bp IncFIA/FIC carrying blaNDM-5 plasmid (isolate 515) was compared to an 172507 IncF plasmid cultured from isolate 85.”

Line 582 – “172, 507 bp”? The”85 IncF” plasmid also doesn’t have blaNDM-5 either?

NEW RESPONSE: No, 85 IncF plasmid does not contain blaNDM-5 and this is likely a different IncF plasmid identified in the isolate. We apologise for the confusion, when we attempted to re-sequence the isolate for long reads, it appears that isolate 85 dropped the NDM containing plasmid/MGE within the plasmid (we sometimes do see this over

time/freeze thaw stress response and stability of plasmid maintenance). The sequence was generated as part of the repeat library prep run (as we had the quality DNA extracted) to compare other plasmid content. To avoid confusion through the narrative and the article, we have now removed this comparative analysis and figure 5 has been remade in line with other comments too.

ORIGINAL COMMENT: Line 445 – BioProject PRJNA1096457 is now available, but no complete assemblies are listed and at least the three complete (and checked) plasmid sequences should be included/submitted separately to GenBank and the accession nos. listed here.

RESPONSE: We apologize for the inconvenience. Genomics data under PRJNA1096457 should now be available, with the supplementary data revised. The plasmid sequences have been uploaded PV171502 (pIncX3-NDM5-a) and PV171503 (pIncF-NDM5).

NEW COMMENT: PRJNA1096457, which contains 190 sets of Illumina data, PV171502 and PV171503 are now available from GenBank, but at least one example of each distinct plasmid carrying bla_{NDM-5} generated by ONT should also be submitted to GenBank, preferably whole genome assemblies for all isolates. The PV171502 and PV171503 entries do not state which isolate they were from or have a link to PRJNA1096457.

NEW RESPONSE: We have updated the data available in PRJNA1096457. Illumina short reads for the 190 have been deposited into SRA. The three hybrid assemblies used for variant calling have been uploaded and clarified in our methods – “ST167 (reference isolate 2525, genome accession JBMWUF000000000), ST1284 (reference isolate 869, JBMWUG000000000) and ST11025 (reference isolate 2623, JBMWUH000000000).” All R10 long-read genomes have been uploaded, which contains all NDM-5 plasmids. Source Data 5 and Source Data 6 have been updated to allow linkage across samples and different sequence data. Our data sharing statement has also been updated accordingly – “**Data sharing**

E. coli Illumina sequence reads, Illumina genomes, R9.4 and Illumina hybrid genomes used for variant calling, R10.4 whole genome assemblies, and R10.4 extracted plasmid sequences used for mapping are available in the NCBI repository under the project accession PRJNA1096457 (Source Data 5-6).”

3) Short-read mapping and Fig. 6

ORIGINAL COMMENT: ...information about plasmid replicons etc in other isolates in the collection is not reported.

RESPONSE: Additionally, we have now mapped the short reads against the two different bla_{NDM-5} carrying plasmids to determine whether we can detect mapped regions and predict whether the IncX3 or IncF plasmid type is present.

NEW COMMENTS:

I would have thought that the most important thing is to try and see if it is possible to work out which plasmid type/variant carries bla_{NDM-5} in each isolate and how this relates to the different ST and sample types (A, H, D). From the text (Lines 170-2) it seems that most isolates have an F replicon (as might be expected for *E. coli*) and many also have an X3 plasmid. From the sequences provided, it seems that bla_{NDM-5} is in different contexts in the two plasmid types. Is this enough to predict which plasmids is likely to carry bla_{NDM-5} in each isolate?

NEW RESPONSE: Thank you for this query. We have restructured our plasmid section in the manuscript to improve the clarity. Yes, our mapping has increased the confidence in the assessment of which plasmid type/variant carried the bla_{NDM-5} in each isolate. Although most

isolates (n = 178) do have an IncF plasmid (as we may expect for *E. coli*), mapping indicates which of these isolates have a greater alignment to the *bla*_{NDM-5} containing region of the plasmid, as indicated by >95% of sequence mapping, and in most cases > 98% (Fig 6).

From our analysis, we suggest that 132 isolates carry the *bla*_{NDM-5} on an IncX3 plasmid and 56 isolates carry the *bla*_{NDM-5} on an IncF plasmid detected. IncF carrying *bla*_{NDM-5} plasmids were mostly (n=53/56) from ST167 isolates (across the phylogeny; not restricted to one clade/branch). We could not elucidate/associate plasmid types to healthcare associated status as there were examples of A, H and D swabs with and *E. coli* carrying *bla*_{NDM-5} gene throughout the time period. This further suggests that these plasmids may be present in the community and within the treatment facility (multiple acquisition routes). Such bacteria are acquired/ patients become colonised over time which may be associated to WASH factors or the use of antibiotics (no control group to evaluate – and therefore not discussed in our study), amongst other possible factors not explored in this observational study.

For 2 isolates, mapping to both plasmid types assessed in the study (PV171502 (pIncX3-NDM5-a) and PV171503 (pIncF-NDM5) was lower than 95% and therefore we are less confident we can resolve which plasmid type is carrying the *bla*_{NDM-5} for those isolates. This may suggest that either an additional plasmid type is present in the cohort but was not detected in our long-read sequencing (which has limitations due to a sub-selection being sequenced), or the original Illumina sequencing was not deep enough provide sufficient depth and coverage against the reference plasmid sequence. We have ensured this is clear in our plasmid section in the results.

Lines 185-6, 186-9 – is this enough to be confident that *bla*_{NDM-5} is on the “pIncF-NDM-5” type or an X3 plasmid, respectively, in these isolates? If so, something could be said about how many isolates carry the gene on each type. Was short-read data from the isolates with R10 ONT data mapped? This might also give useful information about what mapping of data from other isolates means.

NEW RESPONSE: We have added this to our results, and we apologise if this was not clear. As outlined above, from our mapping we suggest that 132 isolates carry the *bla*_{NDM-5} on an IncX3 plasmid, 56 isolates carry the *bla*_{NDM-5} on an IncF plasmid and for 2 isolates, we could not confidently conclude if either plasmid was present. We have also added more information linking isolate ST to plasmid type.

Yes, the short read data was mapped inclusive of isolates with R10 ONT data. The isolate with R10 ONT data to provide PV171503 (pIncF-NDM5) was isolate 25 and 99.9471% of covered bases were mapped. The isolate with R10 ONT data to provide PV171502 (pIncX3-NDM5-a) was 114 and 100% of covered bases were mapped. With these results in mind, we are confident that with our mapping data, especially for mapping % >98% (with equates to n = 177/190 with WGS data) we can predict which plasmid types are most likely to be carrying the *bla*_{NDM-5} gene. The isolates with the lowest % covered bases to either IncF or IncX3 (not including those n = 2 that did not map above 90% to either reference plasmid), were isolates with the lowest sequencing depth, which may suggest a limitation of available sequencing data. Source Data 5 lists all mapping data metrics for each isolate.

Lines 186-193 – this needs to be with the X3 plasmid information, not after the F plasmid information.

NEW RESPONSE: We agree, and the whole plasmid section has been edited and restructured. We discuss the IncX3 data first and then the IncF instead of mixing across the dataset.

Line 190 – the plasmid sequences don't say anything about whether isolates are heterogenous.

NEW RESPONSE: We agree, and this was not our intended meaning. We have amended the sentence: “This varying pattern of mutations among the samples indicates a heterogenous mix of IncX3 plasmids circulating in the *E. coli* population”.

Lines 356-8 – not a complete sentence. You mean one X3 plasmid and one F plasmid, each carrying *bla*_{N_{DM}-5, were concatenated. How much sequence do they share around *bla*_{N_{DM}-5? What effect would this have?}}

NEW RESPONSE: IncX3 plasmid and IncF plasmid (barcode43) were concatenated into a single fasta file. Around 3,000 bp sequence (3 genes) upstream of *bla*_{N_{DM}-5 is shared between both the plasmids (please see Supplementary Figure S4 which highlights this). The concatenation process allowed for more accurate mapping of reads, particularly when competitive mapping was applied.}

Line 358 – what is meant by “competitive mapping”?

NEW RESPONSE: Competitive mapping is a procedure of mapping reads simultaneously against multiple reference genomes. This procedure helps in potentially improving mapping accuracy, detecting contamination in mixed samples and pan-genome alignments. As plasmid backbones can have large repetitive and complex regions (and we predicted multiple potential similar plasmids present in the samples, i.e. IncF), we decided to test two different mapping approaches and compare the data generated. As suggested, the data produced a very similar output (original Fig.6) and therefore we will revise the final figure for the manuscript to display one mapping approach.

We have also clarified this briefly in our methods.

Figure 6

This might be better as supplementary and presented in landscape format – the isolate numbers on the right are upside down. It is also not clear what order these are in – grouping by ST (and for ST167 etc maybe by clade) and indicating these ST might show whether particular plasmid type/SNPs are associated with particular ST etc (see information on Lines 164-6, 179-81). It might also be helpful to indicate with isolates were sequenced by ONT and which plasmid carries *bla*_{N_{DM}-5 in these. The “Competitive” and “Individual” columns look pretty similar, so it may not be necessary to show both. I think that showing the SNPs for the IncF plasmids with large numbers of SNPs is not useful. a)- f) at the bottom and information on this in the legend are not really needed, as each column has a heading. The legend should instead describe what “mean mapping quality” means and what “complex”, “del”, “ins” and “mnp” mean.}

NEW RESPONSE: We have edited Figure 6 to allow the addition of these suggestions. We agree about the display of the large number of mutations and to avoid confusion/overstating results that are not relevant to our specific aim (i.e. to determine whether a particular IncF or an IncX3 plasmid carrying a *bla*_{N_{DM}} gene is present), we have removed the IncF mutations. As correctly summarised by the reviewer, *E. coli* would be expected to carry perhaps multiple different IncF plasmids and therefore it was likely that with a presence of similar sequences in the DNA pool of the whole genome, mismatching or mis-mapping could have occurred,

with the mapping approach aligning a ‘best case’ scenario (similar regions may have passed a threshold, but as the mapping to the plasmid reference sequence for those regions with the large number of mutations was only ~50% of the total sequence it does certainly suggest that it was likely not present in the *E. coli* isolate). Thank you for the suggestion. With figure 6, we did however decide to continue displaying the IncX3 mutations because the plasmid sequences mapped displayed two dominant mutations (further described in other comments) and this provided cohesion across other areas of the manuscript (text and figure 5).

We would therefore prefer to keep this as a main (but revised) figure in the manuscript. We believe the suggested edits to this paper complement other results and provide a cohesive summary for our study. The legend now defines any terminology used, i.e. snp.

4) “admission swab” “hospitalisation swab” and “patient deterioration”

PREVIOUS COMMENT: Line 113 – not clear what is meant by “hospitalisation (patient deterioration)” here. Also Fig. 2.

RESPONSE: We use the term “hospitalisation” in this study to refer to the time period where the children were admitted to the SAM treatment facility (supported by MSF) for treatment and antibiotic therapy as required. The results section has been rewritten following additional advice from co-authors for clarity & to reduce any redundancy between the text and figures – lines 89-101.

PREVIOUS COMMENT: Line 292 – there is nothing in the Methods or Results about how this deterioration was evaluated.

RESPONSE: We apologise for this oversight. We have added “Rectal swabs were also collected at the time of patient deterioration, which was informed by clinical symptoms guided by clinical discretion” within the Study Design section of the methods. – lines 269-270.

PREVIOUS COMMENT: Line 328 – what is meant by “time of deterioration”?

RESPONSE: Additional samples were collected from patients when they deteriorated. This was decided based upon clinical presentation, clinical signs and symptoms of physical deterioration. If the patient was not improving from medical treatment additional rectal samples (and blood if required) were collected.

NEW COMMENT: This is still confusing – the term “hospitalisation swabs” needs to be better explained (or changed). Line 273 states that all children providing swabs for the study were “admitted” to the hospital (Line 273), which could be taken as meaning that they were all “hospitalised”, but “hospitalisation swabs” were taken from only 209 of the children. On Lines 275-7 something like “Additional rectal swabs (defined as “hospitalisation swabs”) were taken if clinical symptoms suggested that the patient was deteriorating” could be used, if this is correct, and add something to Line 79. Also see Lines 545-6, which might imply that not all patients were admitted.

NEW RESPONSE: We have now included the definition of use for “hospitalisation swabs” as suggested. This has been amended as suggested in lines 79-81. The methods section has also been amended for clarity, lines 276-80. This has also been amended as suggested in fig.1 legend, lines 547-551.

5) Other scientific points

ORIGINAL COMMENT: Line 66 – what is the definition of multidrug resistant?

RESPONSE: Multidrug resistance is usually defined as bacterial isolates non-susceptible to at least one antibiotic in three or more antibiotic classes to which they were not intrinsically resistant (Magiorakos, 2012). The sentence has been modified by removing beta-lactam to

avoid suggesting that only use of one class of antibiotics can facilitate MDR.

NEW COMMENT: MDR is used several times in the revised manuscript, so the explanation and reference should be added. I can't follow the second part of the response, as the original sentence didn't seem to mention beta-lactams.

NEW RESPONSE: This has now been added to the manuscript. It was appropriate to add this at the first mention, which was in the introduction. See lines 55-57 and amended text – “Broad spectrum antibiotic use may select for intestinal carriage of multidrug resistant (MDR) gut bacteria, predominantly *E. coli* and *K. pneumoniae*, potentially increasing the risk of transmission within and beyond the gut microbial community^{18,19}. MDR is usually defined as non-susceptibility to at least one antibiotic in three or more antibiotic classes to which they were not intrinsically resistant²⁰.”

ORIGINAL COMMENT: Lines 114-20 – did any children with bacteria with a carbapenemase gene on admission have no such bacteria on discharge?

RESPONSE: For 11 children who had with bacteria with a carbapenemase genes on admission were not colonised with bacteria carrying a carbapenemase gene on discharge. This has been added to the first results section, lines 94-96.

NEW COMMENT: The wording of the response does not make sense or vs. the text, which now states that 11/27 children with bla_{NDM} at admission did not have it at discharge, with the numbers being 14/23 for a bla_{OXA-48}-like gene. Given this, the numbers need checking, and the text is repetitive and could be simplified.

NEW RESPONSE: We apologise for the confusion in the previous response – it looks like some text was omitted in the response to reviewer document previously. Yes, there were children who had a rectal microbiota screened positive by PCR for a carbapenemase gene on admission, that had a negative screen on discharge. We have added this sentence to the results, but in our resubmission, we have altered the structure of the paragraph for flow. This section of the results (Acquisition of Gram-negative bacteria carrying β -lactamase genes) has been restructured to simplify the key take home messages.

The part of the section edited for clarity is in relation to the carbapenemase acquisition data and now reads:

“The acquisition of carbapenemase ARGs was similar for both *bla*_{NDM} and *bla*_{OXA-48}-like, with a 900% increase in *bla*_{NDM} carriage (from 27 to 270 cases) and an 885% increase in *bla*_{OXA-48}-like carriage (from 23 to 261 cases) from admission to discharge (Figs. 1-2a). Among the children who carried bacteria with a *bla*_{NDM} gene or a *bla*_{OXA-48}-like gene at admission, not all remained colonised at discharge with n = 11/27 (*bla*_{NDM}) and n = 14/23 (*bla*_{OXA-48}-like), children found to be negative for these genes at discharge.”

ORIGINAL COMMENT: Line 411 – what are these references? Are the sequences available?

RESPONSE: This section has been re-analysed from new sequencing data. All plasmid sequences used for comparative analysis were from NCBI/PLSDB and the accession number listed indicates sequence availability.

NEW COMMENT: Now Lines 337-8. This question was about references for different ST, not plasmids.

NEW RESPONSE: This was a misunderstanding of additional references. We have added a sentence to the methods to explain which reference genome was used for each ST group, and their corresponding accession numbers:

“SNP phylogenies were constructed using high-quality (hybrid short-read and long-read genomes) reference genomes selected per ST of interest; ST167 (reference isolate 2525, genome accession JBMWUF000000000), ST1284 (reference isolate 869, JBMWUG000000000) and ST11025 (reference isolate 2623, JBMWUH000000000). Illumina fastq files were trimmed as described above, ONT sequence reads were trimmed using filtong (v0.2.1)⁵³ and hybrid assemblies were produced using unicycler (v0.4.9)⁵⁴.”

6) Remaining problems with wording/formatting/organization etc.

ORIGINAL COMMENT: Title – “extended-spectrum beta-lactamase and carbapenemase E. coli” does not make sense and a beta symbol should be used.

RESPONSE: We have amended the title to: “Acquisition of Extended-Spectrum β -Lactamase and carbapenemase positive E. coli in hospitalised children with severe acute malnutrition in Maradi, Niger”

NEW COMMENT: This could still be improved. Suggest “Acquisition of Escherichia coli carrying extended-spectrum β -lactamase and/or carbapenemase genes by hospitalised children with severe acute malnutrition in Maradi, Niger”

NEW RESPONSE: Title changed to - **Acquisition of Escherichia coli carrying extended-spectrum β -lactamase and/or carbapenemase genes by hospitalised children with severe acute malnutrition in Maradi, Niger**

ORIGINAL COMMENT: “blaOXA-48-like-positive/carrying isolates” (e.g. Lines 140, 142) is problematic.

RESPONSE: We have reformatted and edited the text to improve the wording throughout the manuscript.

NEW COMMENT: “blaOXA-48-like-positive/carrying isolates” etc is still used in several places (e.g. Lines 101, 109, 114-5, 117, 119) - this needs to be checked and fixed throughout.

NEW RESPONSE: This has now been amended throughout, with each previous use of carrying or positive checked for appropriate usage and amended to (as example) “carrying a *bla*_{NDM-5} gene” or “positive for a *bla*_{NDM-5} gene” where necessary.

PREVIOUS COMMENT: Line 98 – “and/or” here?

RESPONSE: We thank the reviewer for this comment. This sentence is referring to those children who had a bacterial culture positive for both blaCTX-M-15 and blaOXA-48-like ARGs, as opposed to either or. There was a high carriage rate for blaCTX-M-15 (like) in patients with 76% children having at least one rectal swab positive for bacteria carrying the ESBL (blaCTX-M-15 (like)) ARG.

NEW COMMENT: Now Lines 83-4 “Overall, rectal microbiota from 1,688 samples (1,042 patients, 76%), 338 samples (301 patients, 83 22%), and 339 samples (296 patients, 21.6%) carried clinically relevant blaCTX-M-1-group, blaNDM, and blaOXA-48-like genes, respectively”.

This is still not clear – the “respectively” suggests that the number is for each gene on its own, although “and” instead of “or” is confusing. If 338 is swabs with blaCTX-M-15+blaOXA-48 then this needs to be explained properly. Is 339 this the number for a blaNDM gene alone or blaCTX-M-15+blaNDM? Also, it’s probably better to use “carry a blaCTX-M-1 group gene” rather than “carry blaCTX-M-1 group genes” etc, unless there is evidence for a single isolate having more than one copy? This and use of “and”, “or” and “and/or” needs to be carefully checked throughout the manuscript, to make sure that what is

meant is always both correct and clear. e.g. Line 289 – this needs to be “or vancomycin plus ertapenem” not “and”.

NEW RESPONSE: This section has been further edited and expanded for clarity.

The sentence has now been edited to: “Overall, rectal microbiota from 1,688 samples (1,042 patients, 76%) carried a *bla*_{CTX-M-1} group gene, 338 samples (301 patients, 22%) carried a *bla*_{NDM} gene and 339 samples (296 patients, 21.6%) carried a *bla*_{OXA-48}-like genes.” The following sentences further expand on samples that contained a combination of the different ARG screened for.

When we have used and/or in a sentence with data, we have ensured the most appropriate term applies, we have reverted to the use of “or” instead of “and/or” for the sentence:

“Among the children who carried bacteria with a *bla*_{NDM} gene or a *bla*_{OXA-48}-like gene at admission, n = 11/27 (*bla*_{NDM} positive bacteria) and n = 14/23 (*bla*_{OXA-48}-like positive bacteria), were found to be negative for these genes at discharge.”

We have no evidence of multiple *bla*_{CTX-M} gene copies, so thank you for the suggestion, we have used “carry a *bla*_{CTX-M-1} group gene” instead of “carry *bla*_{CTX-M-1} group genes”.

We have changed “and vancomycin plus ertapenem” to “or vancomycin plus ertapenem” and have carefully checked the usage of “and” and “or” throughout as suggested.

ORIGINAL COMMENT: Line 61 - “high rates of carriage of extended-spectrum beta-lactamase genes”

RESPONSE: This has been edited to: “However, high extended-spectrum β-lactamase (ESBL) carriage prevalence may be linked to...” – lines 48-49.

NEW COMMENT “carriage prevalence” is awkward. See suggested wording in original comment.

NEW RESPONSE: changed to – “However, high rates of extended-spectrum β-lactamase (ESBL) carriage has been associated with amoxicillin exposure in outpatient settings for uncomplicated SAM treatment¹⁸.”

ORIGINAL COMMENT: Line 95 – “Two patients had”.

RESPONSE: This sentence has been rewritten to “Two patients had only one rectal swab collected upon admission” – lines 79-80.

NEW COMMENT: This is still a bit unclear – all patients should have had only one swab collected on admission? Maybe “For two patients, only a single swab was collected, in both cases at admission”

NEW RESPONSE. This has been replaced with the suggested sentence. Please see below text if line numbers are still causing comparison issue. “Additionally, 264 swabs were collected during hospital stay (defined herein as ‘hospitalisation swabs’), if clinical symptoms suggested that the patient was deteriorating, with 12 % (n=168) of patients having three swabs in total, and 3% (n=41) having four to seven swabs collected. For two patients, only a single swab was collected, in both cases at admission.”

ORIGINAL COMMENT: Line 106 – “beta-lactamase genes” is usually used.

RESPONSE: The sub-heading has been changed to “Acquisition of Gram-negative bacteria carrying β-lactamase resistance genes” – line 89.

NEW COMMENT: “Beta-lactamase resistance genes” does not make sense – these genes

APR-25

encode a beta-lactamase but give resistance to beta-lactams. “beta-lactamase genes” is simplest.

NEW RESPONSE: Subheading replaced with “**Acquisition of Gram-negative bacteria carrying β -lactamase genes**” as suggested.

ORIGINAL COMMENT: Line 155 – the section title “Rectal *E. coli* genomic diversity with blaNDM” needs rewording.

RESPONSE: The sub-heading has been reworded to: “Genomic diversity of blaNDM carrying *E. coli*”

NEW COMMENT: “Genomic diversity of *E. coli* carrying a blaNDM gene” might be clearer.

NEW RESPONSE: subheading changed as suggested. Please see:

“Genomic diversity of *E. coli* carrying a blaNDM gene

In total, 248 *E. coli* carrying blaNDM were isolated from 3,004 rectal swabs from 229 patients. Only twelve (5%, n=12/248) were recovered from swabs collected on admission, with 12% (n=30/248) recovered from swabs collected during hospitalisation.”

ORIGINAL COMMENT Line 193-4 – poorly worded – something like “...discharge. This may facilitate transmission into the community as patients return home.” might be more accurate.

RESPONSE: this has been added – lines 154-155.

NEW COMMENT: Lines 155-6? I think that the wording “a high likelihood of community transmission” is too strong – unless you can cite paper(s)/data to support this.

NEW RESPONSE: Thank you for identifying this. We have amended the wording to:

“*E. coli* ST167 with blaNDM-5

Among 148 *E. coli* ST167 isolates recovered from 144 different children, eight (5%) were collected at admission, 19 (13%) during hospitalisation, and 120 (82%) at discharge. The high presence of *E. coli* ST167 carrying blaNDM-5 at patient discharge from the treatment facility may cause concern for ongoing community transmission as patients return home”

ORIGINAL COMMENT: Lines 195-6 – clade A contains clade A?

RESPONSE: This has been edited for clarity. The tree contains two main branches, of which has been split into clades (sub-clades) A-H on one branch with clade I forming a separate more distant lineage – lines 155-160.

NEW COMMENT: The wording here is still difficult to follow. Suggest something like “...*E. coli* ST167 population separated by ~450 SNPs (Fig. 4). One branch contains multiple small subclades (A-G) and one large subclade (H, 80 isolates collected December 2016-July 2017; ≤ 20 SNPs) and the other only a single clade (clade I ≤ 20 SNPs)”, then discuss blaCTX-M-15 and blaCTX-M-55. Specify which blaTEM gene.

RESPONSE: This sentence has been amended as suggested, see lines 168-172.

Lines 161-2 –confusing – what is meant by a “strain” here as opposed to the subclades?

RESPONSE: We apologise for any confusion. This section has now been edited to reflect our original intended meaning (>1 lineage of ST167 *E. coli* present & local divergence of a dominant MDR strain may be occurring):

“The presence of multiple smaller subclades (clades A-G) alongside a larger subclade H (Fig. 4) may suggest evidence of local divergence. The most distant *E. coli* ST167 isolates (clade I, Fig. 4) were within five SNPs of each other, and all carried blaCTX-M-55, whereas the majority

of *E. coli* ST167 (subclades A-H) carried *bla*_{CTX-M-15}. Additionally, isolates within clade I did not co-carry either *bla*_{TEM-1} or *bla*_{OXA-1} genes, unlike those in the other subclades (A-G; Fig. 4). Although multiple patients were colonised with a genetically similar strain of *E. coli* ST167 carrying a *bla*_{NDM-5} gene (subclades A-G, H), two co-occurring *E. coli* ST167 lineages were present in this cohort (clade I)”.

ORIGINAL COMMENT: Lines 337-9 – there should be a space between numbers and units.

RESPONSE: This has been corrected throughout.

NEW COMMENT: Still no space in e.g. Lines 174, 177, 179, 350, 356, 579, 581.

NEW RESPONSE: The line numbers do not seem to match up well, however all numbers and units and all cases of n= (except a % as not a unit/per journal guidelines), now have a space. We apologise for the oversight in the formatting of the text during resubmission. We have used the journal guidelines and recent article publications to ensure our submission has been edited appropriately.

ORIGINAL COMMENT: Line 346 – “from samples positive for a *bla*_{NDM}, *bla*_{KPC} or *bla*_{OXA-48} like gene”.

RESPONSE: This sentence has been changed to: “All phenotypically distinct bacterial colonies were purified from samples positive for a *bla*_{NDM}, *bla*_{KPC} and *bla*_{OXA-48}-like gene for repeat multiplex-PCR” – line 292

NEW COMMENT: No *bla*_{KPC} genes were detected? Line 289 (not 346) – needs to be “or” or “and/or” here, not “and”.

NEW RESPONSE: Amended accordingly. *bla*_{KPC} removed and, the ‘and’ has been changed to and/or.

7) NEW COMMENTS - Discussion

Information is still not well organised and the wording could be improved in places.

Lines 196-8, 199-102 – this seems more like Introduction.

Line 199 – “treatment for complicated SAM” This has been amended to the suggested sentence – “This prospective longitudinal study highlights the high prevalence and acquisition of β -lactamase genes during treatment for complicated SAM”

Line 204 “a globally disseminated” This has been amended to “Our study found a high rate of rectal acquisition of the globally prevalent high-risk *E. coli* strain ST167 carrying *bla*_{NDM-5} which was located on a globally disseminated^{25,26} conjugative IncX3 plasmid.”

Line 212 – where is this mentioned in Results?

NEW RESPONSE: 76% of children were colonised with bacteria carrying *bla*_{CTX-M-1} group, with 47% of children colonised with bacteria carrying *bla*_{CTX-M-1} group at admission. Please see – “Overall, rectal microbiota from 1,688 samples (1,042 patients, 76%), 338 samples (301 patients, 22%), and 339 samples (296 patients, 21.6%) carried clinically relevant *bla*_{CTX-M-1} group, *bla*_{NDM}, and *bla*_{OXA-48}-like genes, respectively”

In the discussion (resubmission v1 line 212) we have rephrased “at baseline” for consistency and used “at admission” – “A decade later, our study confirms a high prevalence of *bla*_{CTX-M-1} group carrying bacteria, with the additional concern: the acquisition of MDR *Enterobacteriales* carrying *bla*_{NDM-5}.”.

Line 216 – “ARG carriage”?

NEW RESPONSE: We agree that this should be used instead of MDR. We have amended this to – “Previous longitudinal studies have shown that carriage with bacteria harbouring ARG is often transient^{18,34}.”

Line 219 – bacteria carrying blaCTX-M-15

NEW RESPONSE: This has been changed to the suggested phrasing – “A decade later, our study confirms a high prevalence of bacteria carrying a *bla*_{CTX-M-1} group gene”. Lines X-X

Line 220 – “an additional concern”

NEW RESPONSE: the additional concern was changed to “an additional concern” as per suggested.

Line 222 – “and suggests” does not make sense here.

NEW RESPONSE: This has been changed for clarification.

“In this study we observed three predominant MDR *E. coli* strains colonising children receiving treatment for SAM. MDR *E. coli* colonisation was noticeably higher at discharge, suggesting nosocomial acquisition and local transmission events.”

Line 224 – “increased carriage of MDR bacteria”

NEW RESPONSE: This sentence has already been amended in line with previous suggestions in this rebuttal.

“Previous longitudinal studies have shown that carriage with bacteria harbouring ARG is often transient^{18,34}”.

Line 226 – where is this in the Results?

NEW RESPONSE: We are citing a previously published manuscript from this cohort on blood stream infections. In line with other comments in this rebuttal, we have clarified the citing of this article and added the citation earlier in our manuscript.

Line 227 – “multiple strains circulating among”

NEW RESPONSE: Line 242, this has been amended from “multiple circulating strains among” to “multiple strains circulating among” as suggested.

Line 231 – Additionally, 31 children... microbiota, were” Where is this in Results? Or cite Ref 20 earlier.

NEW RESPONSE: We have cited reference 20 (now reference 21) earlier – “Additionally, for 31 children in this study carrying *Enterobacteriales* with *bla*_{NDM} in their rectal microbiota, they were further colonised with *Enterobacteriales* species in the bloodstream²¹.”

Line 235 - “received hospitalisation” does not make sense - “are hospitalised”?

NEW RESPONSE: The sentence has been rephrased – “As part of standard care, children presenting with complicated SAM are hospitalised for treatment including antibiotics, until stable.”

Lines 238-9 – wording does not make sense.

NEW RESPONSE: The sentence has been rephrased – “As all children receive antibiotics as mandated by guidance, we were not able to evaluate associations between AMR bacteria acquisition, antibiotic use, or to differentiate other contributing factors such as IPC.”

Line 248 – “factors such as IPC?” meant here. Has this abbreviation been used before? IPC is defined in the introduction, line 59. Thank you for identifying this typographic error, this has now been amended to “and it was not therefore possible to differentiate other contributing factors such as IPC.”

Line 251 – first phrase redundant? What “underscores”?

NEW RESPONSE: This has been rephrased to – “This study underscores the urgent need to strengthen IPC measures in LMIC treatment facilities.”

Line 258 – “E. coli carrying bla_NDM-5”

NEW RESPONSE: The sentence has been amended to - “regarding the dissemination *E. coli* carrying of *bla*_NDM-5 in Madarounfa District Hospital, Maradi”

Line 259 – what is meant by “mirrored”? specifically bla_NDM-5 or different genes?

NEW RESPONSE: This has been changed to – “Unfortunately, Enterobacterales carrying a carbapenemase gene is likely apparent in similar healthcare settings across central and sub-Saharan Africa.”

8) NEW COMMENTS: Methods

Lines 267-283 - order of information could be more logical and reference to complete study details in the previous version seem to have been lost.

NEW RESPONSE: This section has been edited for clarity and to include the original study reference – “Children admitted to the CRENI between 0 and 59 months of age who did not require immediate resuscitation on admission were enrolled²¹. Rectal swabs were collected according to three categories; (1) upon admission to the treatment facility (2) during inpatient stay, additional rectal swabs (defined as “hospitalisation swabs”) were taken if clinical symptoms suggested that the patient was deteriorating, (3) at the time of patient discharge from the treatment facility. All rectal swabs were collected with Amies plus charcoal media (COPAN Diagnostics)) and according to standard anorectal specimen collection guidelines.”

Line 274 – make it clear that only rectal swabs were studied here.

NEW RESPONSE: The sentence has been clarified – “All ethical approvals were relevant for the study and permitted the enrolment of children, collection of rectal samples and analysis for research purposes.”

Line 277-6 – repeat the same information.

NEW RESPONSE: The sentences were revised according to editorial policy, however we have edited slightly to minimise repetition “Ethical evaluation study was approved in 2016 by the National Consultative Ethics Committee of Niger (N007/2016/CCNE) and the Committee for the Protection of Persons, Ile-de-France (16007, Epicentre 2016). All ethical approvals were relevant for the study and permitted the enrolment of children, collection of rectal samples and analysis for research purposes.”

Line 303 – “samples positive for a bla_{CTX-M-1} group gene, we did not look further at isolate with one of these genes.”?

NEW RESPONSE – This sentence has been edited to “Due to the high prevalence of *bla*_{CTX-M-1} group in and likelihood of multiple *Enterobacteriales* species carrying a *bla*_{CTX-M-1} group gene, we did not proceed to isolate bacterial species.”

Line 305 “as described in” is not needed.

NEW RESPONSE: This has been removed and the sentence is now – “The minimum inhibitory concentration of 15 antibiotics were determined for 217 isolates via agar dilution³⁵ and interpreted according to the European Committee on Antimicrobial Susceptibility Testing breakpoints (EUCAST, v13)³⁷”

Line 310 – which kit?

NEW RESPONSE: - The kit used was the QIAamp DNA kit with modification defined in the sentence – “Total gDNA was extracted from pelleted bacteria using QIAamp DNA kit (Qiagen) on a QIAcube instrument (Qiagen, Germany) with an additional RNase step and quantified using the Qubit (v3.0) (ThermoFisher, USA).”

Line 323 – this not a complete sentence.

NEW RESPONSE: The sentence “Sequence read QC and assembly into contigs as described^{35,38}.” Has been removed. The following section clearly details the QC and assembly parameters.

Line 331 – “multi-locus sequence types were determined in silico”

NEW RESPONSE: The sentence has been edited as suggested to – “Multi locus sequence types (MLST) were determined *in silico* using mlst (v2.22.0)⁴⁷.”

Line 332 – “not assigned to a known ST”

NEW RESPONSE: The sentence has been edited as suggested to – “Genomes not assigned to a known ST were uploaded to Enterobase⁴⁸ for assignment of novel allele profiles.”

Line 334 – “Plasmid replicons were detected”

NEW RESPONSE: The sentence has been edited as suggested to – “Plasmid replicons were determined using PlasmidFinder within ABRicate⁵⁰.”

Line 336 – this is not a complete sentence.

NEW RESPONSE: This sentence has been amended to – “SNP phylogenies were constructed using high-quality (hybrid short-read and long-read genomes) reference genomes selected per ST.”

Lines 341-2 – wording does not make sense.

NEW RESPONSE: The methods section on SNP/variant calling has been edited for clarity – “Each reference genome was assembled with unicycler (v0.4.9)⁵², using trimmed short reads as described above. Filtlong (v0.2.1)⁵¹ was used to trim long sequence reads. Snippy (v4.6.0)⁵³ at default parameters was used to call variants against sequence reads, and to generate a SNP phylogenetic alignment of all SNP sites present in the samples. Snippy-clean prepared the resulting SNP alignment for input to recombination removal and phylogenetic tree construction. Gubbins (v2.3.4)⁵⁴ was used to remove possible recombination sites and snp-sites (v2.5.1)⁵³ was used to extract SNPs from the multi-fasta alignment.”

9) NEW COMMENTS: Other minor wording etc

Line 21 – suggest “carriage of Gram-negative bacteria in”

APR-25

Line 23 – “while in hospital”?

NEW RESPONSE: This has been amended as suggested to - “Here, we investigate carriage of Gram-negative bacteria in children under five receiving treatment for SAM in Niger, comparing the frequency of colonization with bacteria carrying resistance genes (ARGs) at admission, during hospital stay and at discharge”.

Line 23 - “carrying a blaNDM gene”? Can blaNDM-5 be specified here? – This has been amended to: “*E. coli* isolates carrying a *bla*_{NDM-5} gene were selected for whole-genome sequencing”

Line 25 – “a blaCTX-M-1 group gene ... a blaNDM gene”? Are all of these known to be blaNDM-5? If so, then “carrying blaNDM-5”. This has been amended to: “Rectal colonisation with bacteria carrying β -lactamase genes was high, with 76% (n=1,042/1,371) of children harbouring bacteria carrying a *bla*_{CTXM-1}-group gene and 25% (n=338/1,371) carrying a *bla*_{NDM-5} gene”.

Line 27 – colonised with what? This has now been amended to: “Over two-thirds of children who did not carry bacteria with a carbapenemase gene at admission were colonised with bacteria carrying a carbapenemase gene at discharge (n=503/729, 69%).”

Lines 32-4 – is Ref 12 also relevant here? This reference has been added to the sentence.

Lines 47-8 – wording is awkward. If only one of the antibiotics is given, then “and” here should be “or”.

NEW RESPONSE: This has been amended for clarity. The recommended treatment is a combination therapy, with the vast amount of literature suggesting a combination of a penicillin and an aminoglycoside antibiotic. The use of metronidazole in complicated SAM has been widely discussed, although is not preferential over a penicillin plus gentamicin.

Lines 48-9 – “a high prevalence of carriage of isolates resistant to extended-spectrum beta-lactams (ESBL)” or “with a gene conferring resistance to extended-spectrum beta-lactams (ESBL)”?

NEW RESPONSE: This has been changed to - Here, we demonstrate a high prevalence of carriage of Gram-negative bacteria (GNB) resistant to extended-spectrum beta-lactams (ESBL)/carbapenemase-producing GNB from rectal swabs in children presenting with complicated SAM.

Line 50 – “high rates of acquisition”?

NEW RESPONSE: This has been changed to – “Additionally, high rates of acquisition of ESBL-producing *E. coli* and *K. pneumoniae* have been reported in hospitalised SAM patients receiving broad-spectrum antibiotics, with a median hospital stay of 10 days¹⁹.”

Lines 52-3 – “Use of broad-spectrum antibiotics”.

NEW RESPONSE: This has been changed to – “Use of Broad spectrum antibiotics may select for intestinal carriage of multidrug resistant (MDR) gut bacteria, predominantly *E. coli* and *K. pneumoniae*, potentially increasing the risk of transmission within and beyond the gut microbial community^{18,19}”

Line 58 - “risk for”?

NEW RESPONSE: Of has been changed to for - the risk for transmission within and beyond the gut microbial community^{18,19}. (full sentence above)

Line 61 – “carriage of antimicrobial resistant”.

NEW RESPONSE: This has been changed to – “There is limited data on the carriage of AMR bacteria in children presenting with SAM in Niger.”

Line 65 – “and/or”?

NEW RESPONSE: This has been changed to “Here, we demonstrate a high prevalence of both carriage and acquisition of ESBL and/or carbapenemase-producing Gram-negative bacteria (GNB) from rectal swabs in children presenting with complicated SAM.”

Line 73 – “the antibiotic most commonly prescribed to this cohort”?

NEW RESPONSE: This has been changed to – “The antibiotic most commonly prescribed to this cohort was ceftriaxone, administered both prior to and during hospitalisation as part of the MSF-supported nutritional treatment program in Niger²¹.”

Lines 85-6 – “both a blaCTX-M-1 group gene and a blaNDM gene” etc? Note extra dash in blaCTX-M-1.

NEW RESPONSE: This sentence has been amended to – “Among these, 227 samples from 207 patients carried bacteria both a *bla*CTX-M-1 group gene and a *bla*NDM gene,”
The extra dash has been removed.

Line 92 – “a blaCTX-M-1 group gene”?

NEW RESPONSE: This has been changed to – “At admission, 47% (n=642/1,371) of patients tested positive for at least one of the screened ARGs, with 97% (n=621/642) carrying a *bla*CTX-M-1 group gene (Fig. 1).”

Lines 94 – “a blaOXA-48-like gene”. “A blaNDM and/or a blaOXA-48-like gene”?

NEW RESPONSE: This has been amended to – “Among the children who carried bacteria with a *bla*NDM gene and/or a *bla*OXA-48-like gene at admission,”

Line 98 – “blaCTX-M-1-like genes”

NEW RESPONSE: This has been amended to – “The prevalence of *bla*CTX-M-1-like genes increased from 45% (n=621) at admission to 66% (n=904) at discharge, representing a 45% increase (Fig. 1, Fig. 2a).”

Line 105 – “All rectal swabs in which at least one carbapenemase gene was detected”?

NEW RESPONSE: This has been amended to – “All rectal swabs in which at least one carbapenemase gene was detected were further scrutinised to identify bacterial species.”

Line 109 – you mean that more samples had them?

NEW RESPONSE: Yes, more samples collected at discharge from the treatment facility had bacteria carrying a *bla*NDM gene compared to samples collected at admission and to samples collected during hospitalisation. This sentence has been changed to:

“Notably, *E. coli* positive for a *bla*NDM gene and *K. pneumoniae* positive for a *bla*OXA-48-like gene were detected in the rectal microbiota from samples collected during hospitalisation and at discharge compared to those collected on admission (Fig. 2a). – lines 114-117.

Line 122 – “E. coli carrying a blaNDM gene”

NEW RESPONSE: This has been amended to - “In total, 248 *E. coli* carrying a *bla*_{NDM} gene were isolated from 3,004 rectal swabs from 229 patients.”

Line 126 – “patient” not really needed here.

NEW RESPONSE: The sentence has been amended to – “The remaining 83% (n=206/248) were recovered from swabs collected at discharge from the treatment facility”.

Lines 127-9 – wording can be simplified.

NEW RESPONSE: This sentence has been reworded to – “Between 2019-2022 190 *E. coli* isolates carrying a *bla*_{NDM} gene were sequenced (n=190/248, 77% of the collection) to an average short read sequencing coverage of 30-70X, median 53X (Supplementary Fig. S1).”

Line 138 – “Levels of resistance in *E. coli* carrying a *bla*_{NDM} gene were lowest”

NEW RESPONSE: This sentence has been reworded to “Levels of AMR in *E. coli* carrying a *bla*_{NDM-5} gene were lowest to tigecycline (n=0/217 0%), fosfomycin (n=6/217, 3%) and amikacin (n=77/217, 35%) (Fig. 3a).”

Line 140 – “Rates of resistance to carbapenems were high, as might be expected” Maybe say this first?

NEW RESPONSE: This sentence has been rephrased & moved to the beginning of the section – “**AMR and genetic mechanisms of resistance in *E. coli***

Rates of resistance to carbapenems were high, as might be expected with 61% (n=133/217) and 73% (n=159/217) resistance for imipenem and meropenem, respectively.”

Line 143 – “WGS on n=7/21 confirmed” – Isolates phenotypically sensitive to meropenem did contain *bla*_{NDM-5}, where WGS data was available (the 7 out of 21 isolates sensitive to meropenem had WGS data available). This has been amended to “Of the 21 isolates sensitive to meropenem, where WGS was available (n=7/21), a carbapenemase gene was detected” for clarity.

Line 144 – “a carbapenemase gene”? Which? – both *bla*_{OXA-48} like and *bla*_{NDM}, or just one type? Lines 144-6 – this would be better in the previous section on *E. coli* with *bla*_{NDM}.

NEW RESPONSE: This sentence has been rephrased for clarity – “For all *E. coli* that were subject to WGS, a *bla*_{NDM-5} gene was the only *bla*_{NDM} variant. When *bla*_{OXA-48} gene was identified, (n=2), it was detected alongside a *bla*_{NDM-5} gene. This has now been moved to the previous section as suggested.

10) Figures

Fig. 1

ORIGINAL COMMENT: This is not particularly easy to follow and the information is also in the text. A clearer figure (supplementary?) would mean that most of the detail could be removed from the text.

RESPONSE: We agree that Figure 1 was not particularly easy to follow in the submitted form and there is some duplication with the text in the first results section. The figure has been edited for clarity; however, we still feel this should be Figure 1 in the main text to help set the scene for following dataset.

NEW COMMENT: The wording in most boxes has problems and needs fixing. e.g. “642 children had a rectal sample carrying at least one ARG on admission”. Line 547 “is detailed”.

It might be helpful to what was sequenced on this figure, if it can be done simply, as this gets a bit lost in the text.

NEW RESPONSE: Figure 1 has been redesigned and edited further in this revision. We have edited wording inconsistencies, and the text is now consistent to the main article. We have also added what has been sequenced to link to the text and Fig.1, line 133-134 – “Between 2019-2022 190 *E. coli* isolates carrying a *bla*_{NDM} gene were sequenced (n=190/248, 77% of the collection, Fig.1) to an average short read sequencing coverage of 30-70X, median 53X (Supplementary Fig. S1).”

Fig. 2

Line 552 – can just be “carbapenemase gene” and did each child have only one? Line 555 – fix “others grouped as others” Line 556 – is this all *bla*_{NDM-5} and/or all *bla*_{OXA-48}?

NEW RESPONSE: This has been corrected to – “a) The number of children who were carrying and acquiring bacteria with a carbapenemase gene in their rectal microbiota at admission, during hospitalisation for treatment, and at patient discharge. Data is split according to the two most dominant bacterial species; *E. coli* and *K. pneumoniae*, with the remaining grouped as others (a child with a ≥ 1 carbapenemase gene in different bacteria resulted in a single count per bacteria species).”

Some children did have >1 carbapenemase gene (both a *bla*_{NDM} gene and a *bla*_{OXA-48}-like positive PCR) n=89 from the samples (mixed bacteria). Fig. 2 summarises the bacteria carrying a carbapenemase gene split across both bacterial species and sample collection time. Total counts were provided, therefore, if a *K. pneumoniae* isolate carrying *bla*_{OXA-48}-like and an *E. coli* isolate was cultured carrying *bla*_{NDM}, that would result in two counts on the graph in total.

Lines 91-93 have now been further edited to include that detail “Among these, 227 samples from 207 patients carried bacteria with both a *bla*_{CTXM-1} group gene and a *bla*_{NDM} gene, 189 samples from 178 patients carried bacteria with both a *bla*_{CTXM-1} group gene and a *bla*_{OXA-48}-like gene, 91 samples from 89 patients carried bacteria with both a *bla*_{NDM} gene and a *bla*_{OXA-48}-like gene, and, 51 samples from 51 patients carried bacteria with all three ARG types.”

Fig 3

Three letter abbreviations are more common. e.g. CTX for cefotaxime, as in *bla*_{CTX-M}, CAZ for ceftazidime.

NEW RESPONSE: This figure has been remade with three letter abbreviations for all following the EUCAST guidelines for consistency:

https://www.eucast.org/fileadmin/src/media/PDFs/EUCAST_files/Disk_test_documents/Disk_abbreviations/EUCAST_system_for_antimicrobial_abbreviations.pdf

Fig. 4

A-G called “subclades” in text. Line 570 – reword “blown up... for increased resolution”.

Line 572 – fix “hospitalisation”. Line 573 – “of the patient is indicated (F, female; M, male)”.

Lines 574-6 – suggest “beta-lactamase genes is” See previous comment on “presence/absence rectangle”, “carbapenemase genes in blue, *bla*_{CTX-M} genes in green...” etc.

NEW RESPONSE: The figure legend has been amended to -

“Fig.4 – *E. coli* ST167 SNP phylogenetic tree. The clades within the tree are labelled A-I and are enhanced for visualisation in the purple and blue boxes. Blue boxes within clade H

indicate isolates with the most pairwise SNPs detected within that group. The tree leaves are coloured according to sample type (at admission, during hospital stay, discharge) and named according to patient ID. The sex of the patient is indicated (F, female, M, male), followed by the age of the child in months and the date of the sample (month and year). The presence of acquired β -lactamase antibiotic resistance genes (ARG) is denoted with a presence/absence heatmap with a carbapenemase gene denoted blue, a *bla*_{CTX-M-1} group denoted green and a *bla*_{OXA-1/bla}_{TEM} (variants grouped) denoted brown”.

11) Supplementary

The supplementary data section has improved, but still has some problems.

Fig. 1 – its not clear why this has an extra title at the top. “Displaying the” at the start of the legend is not needed and should “genome depth” be “coverage”?

This extra title has been removed, and the legend amended accordingly.

Fig. S2 and Fig. S3 legends could be simplified and condensed e.g. remove text that simply repeats Methods, all isolates are from rectal swabs, discussing admission samples first seems more logical.

These have been amended accordingly, condensing and reordering.

Fig. S4 - at this scale it is hard to see anything much on these diagrams and most genes are not labelled. Figure 5 in the main manuscript already shows more detail and more plasmids.

This has been deleted.